# Interpretable Dimensionality Reduction by Feature-preserving Manifold Approximation and Projection

## Abstract

Nonlinear dimensionality reduction often lacks interpretability due to the absence of source features in low-dimensional embedding space. We propose FeatureMAP, an interpretable method that preserves source features by tangent space embedding. The core of FeatureMAP is to use local principal component analysis (PCA) to approximate tangent spaces. By leveraging these tangent spaces, FeatureMAP computes gradients to locally reveal feature directions and importance. Additionally, FeatureMAP embeds the tangent spaces into low-dimensional space while preserving alignment between them, providing local gauges for projecting the high-dimensional data points. Unlike UMAP, FeatureMAP employs anisotropic projection to preserve both the manifold structure and the original data density. We apply FeatureMAP to interpreting digit classification, object detection and MNIST adversarial examples, where it effectively distinguishes digits and objects using feature importance and provides explanations for misclassifications in adversarial attacks. We also compare FeatureMAP with other state-of-the-art methods using both local and global metrics.

## 1 Introduction

Nonlinear dimensionality reduction methods are widely used for visualising and preprocessing high-dimensional data in machine learning (Tenenbaum et al., 2000; Roweis & Saul, 2000; Zhang & Wang, 2006; Donoho & Grimes, 2003; Belkin & Niyogi, 2003; Zhang & Zha, 2004; Van der Maaten & Hinton, 2008; McInnes et al., 2018). These methods assume that the intrinsic dimensionality of the underlying manifold is significantly lower than the ambient dimensionality of real-world data (Levina & Bickel, 2004; Pope et al., 2021; Wright & Ma, 2022). Nonlinear dimensionality reduction techniques approximate this manifold using a discrete graph, such as a $k$-nearest neighbor ($k$NN) graph. These methods project data from high-dimensional to low-dimensional space to preserve the topological structure of the original data.

While nonlinear dimensionality reduction is effective for visualizing high-dimensional data, one major limitation is the lack of interpretability in the reduced-dimension results (McInnes et al., 2018). Unlike linear methods such as Principal Component Analysis (PCA), where the dimensions in the embedding space correspond to the directions of the greatest variance of the original data, nonlinear methods do not provide such clear interpretations. Specifically, nonlinear dimensionality reduction, such as t-SNE and UMAP, prioritises preserving distances between data points, which often results in the loss of source feature information in the embedding space. Consequently, it fails to illustrate feature contributions, or loadings, as effectively as linear methods like PCA, making it challenging to explain the significance of individual features in the reduced-dimensional space.

In this paper, we aim to enhance the interpretability of nonlinear dimensionality reduction. Beyond preserving the topological structure of data points in the embedding space, we focus on incorporating source features to create a more interpretable approach. Feature information is encoded within the column space of the data, and we use the tangent space to locally represent this column space (Singer & Wu, 2012; Lim et al., 2021). The concept of employing tangent space arises from the observation of anisotropic density on a manifold, where some curves passing through a point are

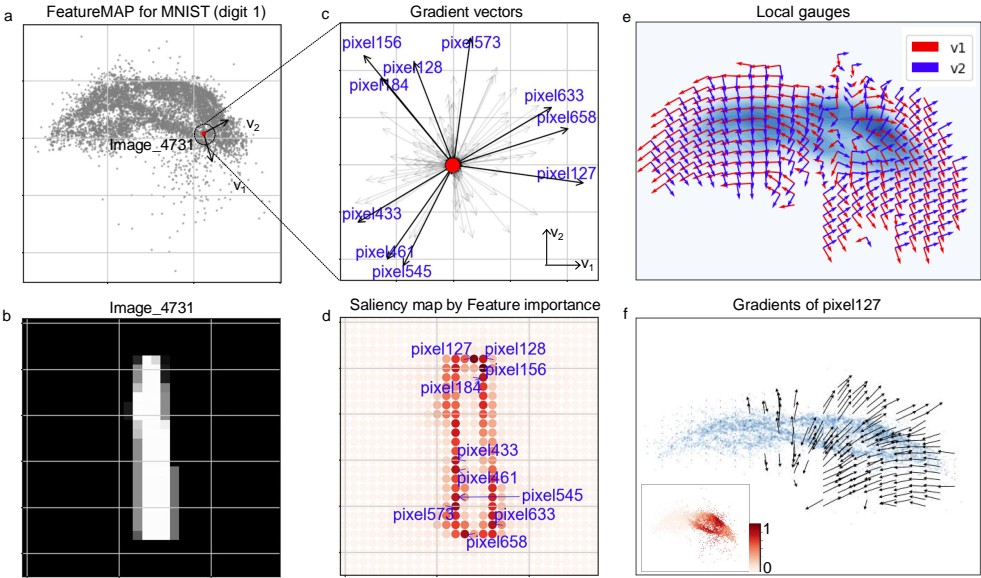

Figure 1: FeatureMAP enables gradients in visualization. a. FeatureMAP embeds the group of digit 1 images from the MNIST dataset into a two-dimensional space. b, c. A randomly selected data point (highlighted in red) is displayed along with its associated tangent space, spanned by the basis vectors $v_1$ and $v_2$, with the top-10 feature gradients annotated. d. The feature importance, derived from the magnitude of the gradients, is mapped onto the selected image as a saliency map. This map highlights the the digit's edge through feature importance. e. The tangent spaces are visualized using local gauges by local PCA bases. f. The feature gradients of pixel 127 are visualized, showing both direction and magnitude (inset plot).

flat while others are steep, reflecting the varying degrees of feature variation in different directions. The set of tangents to these curves forms the tangent spaces, which incorporates the source feature information.

We propose **Feature**-preserving **m**anifold **a**pproximation and **p**rojection (FeatureMAP), to address these gaps by integrating feature preservation directly into the embedding process. Unlike existing methods, it embeds tangent spaces alongside neighborhood graphs, enabling a more faithful representation of local geometry and feature relationships. This distinct approach ensures that the embedding not only maintains global and local accuracy but also supports interpretability by visualizing feature contributions.

We approximate the manifold's topological structure using a $k$NN graph and compute the tangent space by local PCA at the local nearest neighbours. Based on the tangent space, FeatureMAP computes the gradient for each feature, enabling the computation of both feature importance and the direction at each point. Feature importance at a data point serves as a saliency map, highlighting the significant features after embedding, while feature direction illustrates how features change locally in the low-dimensional space ( Fig. 1). Additionally, FeatureMAP embeds the tangent spaces to construct local gauges for projecting data points in the low-dimensional space. The gauge at each point is associated with a hyperellipsoid, with its radii determined by the singular values, reflecting anisotropic density. Thus, FeatureMAP preserves this anisotropic density by maximising the correlation between the local anisotropic radii in the high- and low-dimensional spaces.

To summarize, we make the following contributions:

- We propose FeatureMAP, an interpretable nonlinear dimensionality reduction method that preserves source features and local density.
- We evaluate FeatureMAP on digit and object data. FeatureMAP uses feature gradients to successfully explain the digit classification and object detection.
- We apply FeatureMAP to MNIST adversarial examples to explicitly show that the adversarial attack changes the feature importance, which fools the LeNet classifier.

In the following sections, we begin by discussing the related works before introducing the proposed method. This is followed by experiments demonstrating the method's application to digit classification, object detection, and MNIST adversarial examples. We also present a comparison with state-of-the-art methods using both local and global evaluation metrics.

## 2    RELATED WORK

Nonlinear dimensionality reduction is often considered superior for preserving distance and neighborhood information (Borg & Groenen, 2005; Van der Maaten & Hinton, 2008; McInnes et al., 2018). Methods like t-SNE, UMAP, and LargeVis rely on kNN graphs to approximate the manifold (Van der Maaten & Hinton, 2008; McInnes et al., 2018; Tang et al., 2016). Recent improvements, such as TriMAP(Amid & Warmuth, 2019), PaCMAP (Wang et al., 2021), and DensMAP (Narayan et al., 2021), aim to better preserve global and local properties. Moreover, h-NNE (Sarfraz et al., 2022) applied a hierarchical nearest neighbour graph to preserving multi-level grouping properties of original data. SpaceMAP (Zu & Tao, 2022) used space expansion to match the high and low dimensional space, while CO-SNE (Guo et al., 2022) extended t-SNE from Euclidean space to hyperbolic space. However, these methods lack interpretability as they do not incorporate source features into the embeddings.

The interpretability of nonlinear dimensionality reduction has been largely overlooked in the design and evaluation of embedding methods (Liu et al., 2016; Vellido et al., 2012; Frénay & Dumas, 2016; Dumas et al., 2018). While linear methods like PCA naturally provide interpretability by revealing source features (Gabriel, 1971), few approaches have addressed this issue in nonlinear methods. Liu *et al.* (Liu et al., 2016) examined the trade-off between interpretability and embedding structure, and Bibai *et al.* (Bibal & Frénay, 2019; Bibal et al., 2018; Marion et al., 2019; Sips et al., 2009) proposed explaining low-dimensional axes and scatter plot positions. Wu *et al.* (Wu et al., 2019) introduced subspace projection for kernel dimensionality reduction, and Bibal *et al.* (Bibal et al., 2020) adapted local interpretable model-agnostic explanations (LIME) to explain t-SNE. Bardos *et al.* (Bardos et al., 2022) introduced a model-agnostic explanation technique for dimensionality reduction. More recently, Corbugy et al. (2024) proposed gradient-based explanations for nonlinear methods, relying on loss function's derivatives. In contrast, our method, FeatureMAP, directly computes gradients over the manifold using local PCA, enhancing the exploration of topological structures and enabling a more interpretable embedding. Additionally, the debate on UMAP and t-SNE revolves around interpretability, with critiques like Chari & Pachter (2023) arguing their embeddings can be arbitrary, while others Lause et al. (2024) demonstrate their potential for biologically meaningful insights despite not preserving high-dimensional distances. FeatureMAP builds on manifold learning by not only preserving the topological structure but also retaining source features in the tangent space, enhancing the interpretability through local feature gradient analysis within the embedded tangent space.

## 3    THE FEATUREMAP METHOD

We present FeatureMAP to augment manifold approximation by preserving feature information through gradient calculation. We begin by computing the tangent spaces by local PCA, which enables computing manifold gradients. Next, we embed the tangent spaces while preserving their alignment, and then project the data points along the embedded gauges into a low-dimensional space. In the following sections, we elaborate on each step and validate the proposed methods. Fig. 2 illustrates the framework of our method.

### 3.1    MANIFOLD GRADIENTS

We follow the manifold assumption, where the data points $\{x_1, ..., x_m\}$[1] $\subset \mathbb{R}^n$ lie on a $d$-dimensional Riemannian manifold $\mathcal{M}^d$ embedded in $\mathbb{R}^n$, with the intrinsic dimension $d \ll n$. For each data point $x_i$, the $\mathcal{M}^d$ has a tangent space $T_{x_i}\mathcal{M}$ consisting of all vectors at $x_i$ that are tangent to the manifold. Let $\{f_1, ..., f_n\}$ denote the basis of the data points, which spans the feature space.

---

[1]Vectors of lower-case letters refer to column vectors.

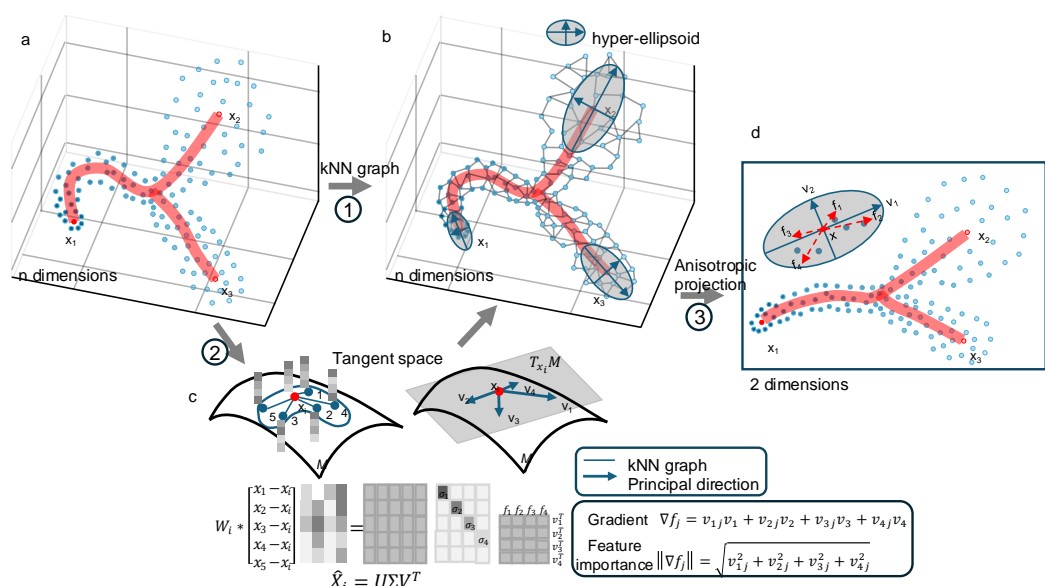

Figure 2: The FeatureMAP framework. a, b, c. FeatureMAP begins by taking $n$-dimensional data as input. In step 1, it constructs a topological space using a $k$-nearest neighbors (kNN) graph. In step 2, it computes the tangent space and corresponding gradients using local PCA, and then embeds the tangent spaces to preserve their alignment (as shown in Figure Fig. 3). d. In step 3, along the embedded tangent space, FeatureMAP applies an anisotropic projection to project the data into a low-dimensional space, while illustrating the feature gradients in the local gauges.

We first depict the manifold structure by a weighted $k$NN graph (Fig. 2) which approximates the geodesic distance on the manifold (Dong et al., 2011). Let $(X, E)$ denote the $k$NN graph, where $X = \{x_1, ..., x_m\}$ are the data points and $E$ the set of edges $(i, j)$. The edge weight $P_{ij}$ of the $k$NN graph is calculated by the probability distribution on original data, which is same as (Dong et al., 2011; McInnes et al., 2018) as

$$\tilde{P}_{j|i} = \exp(-(\|x_i - x_j\| - dist_i)/\gamma_i)$$
$$P_{ij} = \tilde{P}_{j|i} + \tilde{P}_{i|j} - \tilde{P}_{j|i}\tilde{P}_{i|j} \tag{1}$$

where $x_j$ is within the $k$ nearest neighbours of $x_i$, $dist_i$ is the distance from $x_i$ to its nearest neighbour and $\gamma_i$ is adaptive for each $i$ corresponding to the length-scale. Note that the approximate kNN algorithm (Dong et al., 2011) provides theoretical guarantees and has been experimentally shown to effectively retrieve true positives while controlling false positives, even in the presence of noise.

Based on the $k$NN graph depicting the topological structure, we retrieve the important features of the data points $X$. Obviously, the feature information is encoded in the column space of $X$, where the principal components capture the majority of information (Jolliffe & Cadima, 2016). Meanwhile, the features are not evenly distributed on the manifold because of curvature: some area on manifold presents plane surface, while some bends more sharply. This topological heterogeneity suggests us to locally extract important features by the local principal component analysis (PCA).

Given a data point $x_i$, we center its $k$ nearest neighbours as $X_i = [x_i^{(1)} - x_i, ..., x_i^{(k)} - x_i]^T \in \mathbb{R}^{k \times n}$. Note that data points are locally modeled as a weighted $k$NN graph. Considering the edge weight $P_{ij}$ of $k$NN graph in Eq. (1), we construct the weight matrix as $W_i = diag(\sqrt{P_{ii_1}}, ..., \sqrt{P_{ii_k}}) \in \mathbb{R}^{k \times k}$. Thus, larger weights are assigned to closer neighbours to emphasize the local neighborhood of each data point $x_i$, ensuring that the most significant local variation is retained.

To locally derive the principal components around data point $x_i$, we apply weighted singular value decomposition (SVD) [2] to $X_i$ and get

$$\hat{X}_i = U_i \Sigma_i V_i^T \tag{2}$$

---

[2]In practice, $k$ is constant (e.g., 15), thus the extra time cost for SVD is $O(m)$.

where $\hat{X}_i = W_i X_i$. The singular values $\Sigma_i = diag(\sigma_{i1}, ..., \sigma_{ik})$ are in decreasing order and the corresponding right eigenvectors $V_i = [v_1, ..., v_k]$ span the column space of $\hat{X}_i$ (Fig. 2). We show that the local PCA space $V_i$ approximates the tangent space $T_{x_i}\mathcal{M}$ by the following theorem.

**Theorem 3.1 (Singer & Wu (2012); Lim et al. (2021))** *The PCA basis $V_i = [v_1, ..., v_d]$ by weighted SVD approximately represents an orthonormal basis for the tangent space $T_{x_i}\mathcal{M}$.*

We keep the largest $d$ singular values and corresponding right eigenvectors. The intrinsic dimension $d_i$ around data point $x_i$ is locally estimated as the number of singular values that captures most of the data's variability, with $d$ set as the median of $\{d_i | i = 1, ..., m\}$.

The tangent space represents all possible directions in which one can move from a point $x_i$ on the manifold. Among these directions, the gradient vector identifies both the direction and rate of steepest ascent of a function on the manifold, aiding in the interpretation of the local space around point $x_i$. Next, we show how to compute the gradient vector at $x_i$ based on the approximate tangent space.

Consider a function $f : \mathcal{M} \to \mathbb{R}^n$, mapping the manifold to its ambient space. Let $f_j(x) = x_j$ represent the $j$-th feature. The gradient of $f_j$ is $\nabla f_{j|x} = e_j$ where $e_j$ is the standard basis vector. By projecting this gradient to the tangent space, we obtain $c_i = \langle \nabla f_{j|x}, v_i \rangle = v_{ij}$. Thus, the gradient vector of function $f_j$ on the manifold is

$$\nabla f_j = \sum_{i=1}^{d} c_i v_i = \sum_{i=1}^{d} v_{ij} v_i. \tag{3}$$

This gradient vector depicts the change of $j$-th feature. The vector $\nabla f_j$ points in the direction of the steepest increase for the feature $f_j$ (Fig. 1c,f), and its magnitude, $||\nabla f_j||$, corresponds to the rate of increase in that direction. The magnitudes of the gradient provide a saliency map for ranking features (Fig. 1), where larger magnitudes indicate more important features. We define feature importance based on the gradient magnitude as follows.

**Definition 3.1 (Feature importance)** *The feature importance at a data point $x$ is defined as the magnitude of its gradient by*

$$||\nabla f_j|| = \left(\sum_{i=1}^{d} |v_{ij}|^2\right)^{\frac{1}{2}}, \; j = 1, ..., n. \tag{4}$$

The feature importance characterises the increase rate of the $j$-th feature at the data point $x$. Higher feature importance indicates greater variability at this point, providing a local explanation for the differences in the surrounding data. For example, as shown in Fig. 1d, the saliency map based on feature importance highlights the edge patterns in the image. This occurs because the pixel features along the edge changes more rapidly than those in other areas when the given image is compared to its local neighbors. In addition to feature importance, we also show the directions of feature increase at a single data point and across the manifold embedding (Fig. 1c, f). These dynamic patterns by gradients explain how the feature is changing at each data point and along the manifold.

It is important to note that the tangent space facilitates gradient calculation, enabling us to interpret the underlying features. In contrast, conventional nonlinear dimensionality reduction methods, such as diffusion maps, t-SNE, and UMAP (Coifman et al., 2005; Van der Maaten & Hinton, 2008; McInnes et al., 2018), do not account for the tangent space, resulting in a lack of interpretability in the low-dimensional embedding space.

## 3.2 TANGENT SPACE ALIGNMENT

The tangent space approximation via local PCA serves as local gauges for data points, varying smoothly across the manifold. Before projecting the high-dimensional data points into a low-dimensional space, we first project the gauges into the low-dimensional space. Similar to the pairwise distance between data points, we consider the pairwise relationship of tangent spaces by their alignment (Fig. 3). We then embed the tangent spaces by maintaining the pairwise alignment.

For two close data points $x_i$ and $x_j$, the tangent spaces are connected by parallel transport, which is estimated using the rotation matrix between their respective local PCA bases (Singer & Wu, 2012).

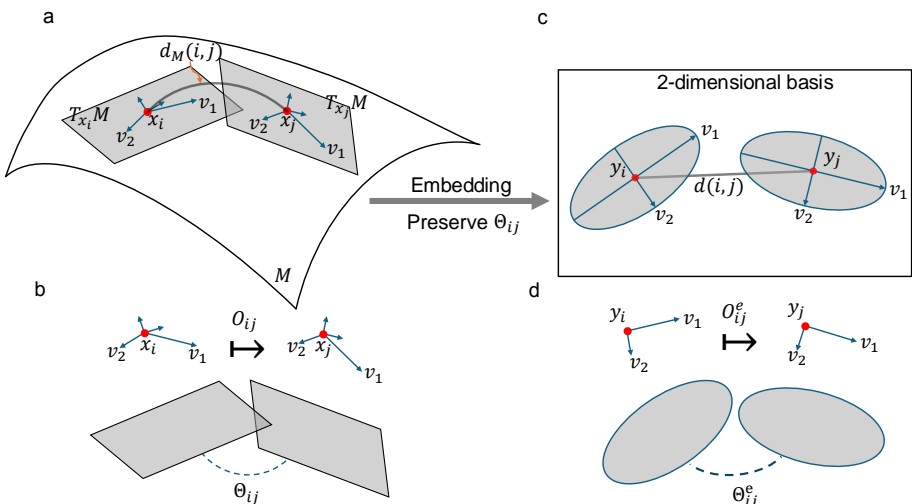

Figure 3: Tangent space alignment and embedding. a. The tangent spaces at data points $x_i$ and $x_j$ are associated with basis vectors (with $v_1$ and $v_2$ annotated) in high-dimensional space. b. The alignment from point $i$ to point $j$ involves a rotation $O_{ij}$ with angle $\Theta_{ij}$. c, d. The tangent space embedding is computed by preserving the rotation angle $\Theta_{ij}$.

We define this rotation matrix as the alignment between the two tangent spaces. Formally, given two data points $x_i$ and $x_j$ with distance $d_{\mathcal{M}}(i,j)$ on the manifold (Fig. 3), the orthonormal matrices $V_i, V_j$ from local PCA act as the bases of tangent spaces $T_{x_i}\mathcal{M}$ and $T_{x_j}\mathcal{M}$, respectively. The alignment between the two tangent spaces is the rotation matrix by the following optimal alignment $O_{ij}$:

$$O_{ij} = \arg \min_{O \in O(d)} \|O - V_j V_i^T\|_F. \tag{5}$$

Numerically, the alignment of orthonormal matrix $O_{ij}$ is computed by the SVD of $V_j V_i^T = U'\Sigma'V'^T$ and $O_{ij} = U'V'^T$.

In addition, we define the transformation between $x_i$ and $x_j$ as:

$$\Gamma_{i \to j} : X \mapsto O_{ij}X + d_{\mathcal{M}}(i,j). \tag{6}$$

The transformation $\Gamma_{i \to j}$ represents the connection between data points $i$ and $j$ in both topological space (through distance $d_{\mathcal{M}}(i,j)$) and tangent space (through alignment $O_{ij}$).

Our goal is to preserve both the topological structure and the tangent spaces to enable feature interpretation in low-dimensional space. We first embed the tangent spaces by preserving their alignment (Eq. (5)) to create local gauges, and then project the data points along these embedded gauges to maintain the local distances in a low-dimensional space.

### 3.3 TANGENT SPACE EMBEDDING

In this section, we demonstrate how to project the tangent spaces into a low-dimensional space by preserving their alignment (Fig. 3).

For each point $x_i$, the tangent space $T_{x_i}\mathcal{M}$ is estimated by the $d$ largest right eigenvectors $V_i = [v_1, ..., v_d]$ in Eq. (2). We define the similarity between the tangent spaces $V_i$ and $V_j$ as

$$\cos \Theta_{ij} = \frac{\langle V_i, V_j \rangle_F}{\|V_i\|_F \|V_j\|_F} \tag{7}$$

where $\langle V_i, V_j \rangle_F$ denotes the Frobenius inner product and $\|\cdot\|_F$ is Frobenius norm. Eq. (7) presents the cosine similarity between tangent space $V_i$ and $V_j$, which induces a general angle $\Theta_{ij}$ as shown in Fig. 3. Note that this similarity quantitatively characterises the alignment between two tangent

spaces (Eq. (5)). Based on the similarity, we define the probability distribution on the tangent spaces $V = \{V_1, ..., V_m\}$ in terms of cosine distance as

$$\mathbf{S}_{j|i}(\Theta) = \exp(-(|1 - \cos\Theta_{ij}| - dist_i')/\gamma_i') \tag{8}$$

where $1 - \cos\Theta_{ij}$ is the cosine distance of these two tangent spaces, $\gamma_i'$ is related to the local distance and $dist_i'$ is the smallest distance from $x_i$ to its neighbours. $\mathbf{S}_{j|i}(\Theta)$ follows a rescaled exponential distribution similar to Eq. (1), with normalization applied in the same way as in Eq. (1). In the following section, we focus on preserving the pairwise cosine distance between tangent spaces in the low-dimensional embedding space.

Consider projecting the manifold into a $d'$-dimensional space ($d' \leq d \ll n$). For each data point $x_i$, let the embedding tangent space be $V_i^e = [v_1^e, ..., v_{d'}^e] \in \mathbb{R}^{d' \times d'}$. Similarly, we define the cosine similarity between $V_i^e$ and $V_j^e$ as

$$\cos\Theta_{ij}^e = \frac{\langle V_i^e, V_j^e \rangle_F}{\|V_i^e\|_F \|V_j^e\|_F} \tag{9}$$

where $\Theta_{ij}^e$ is the angle between the embedded tangent space. The probability distribution on the tangent space embedding, which inherits the heavy-tailed distribution from (Van der Maaten & Hinton, 2008; McInnes et al., 2018) as:

$$\mathbf{T}_{ij}(\Theta^e) = (1 + ad_{ij}^{2b}(s,t))^{-1} \tag{10}$$

where $d_{ij} = 1 - \cos\Theta_{ij}^e$ represents the cosine distance between the embedded tangent spaces $V_i^e$ and $V_i^e$ and $a, b$ are shape parameters.

Our goal is to compute the tangent space embedding $V^e = \{V_1^e, ..., V_m^e\}$ in a way that maximizes the agreement between the original and embedded tangent spaces by matching the probability distributions of cosine distances. To achieve this, we minimize the difference between the probability distributions $\mathbf{S}$ and $\mathbf{T}$ by Kullback–Leibler divergence:

$$KL(\mathbf{S}\|\mathbf{T}) = \sum_{ij} \mathbf{S}_{ij}(\log\mathbf{S}_{ij} - \log\mathbf{T}_{ij}). \tag{11}$$

The tangent space embedding $V^e = \{V_1^e, ..., V_m^e\}$ provides local gauges for visualizing feature gradients (Fig. 1e, 2d). These local gauges $V^e$ represent the principal directions along which the data points are locally distributed in the low-dimensional embedding space. The directions of the gauge $V_i$ are weighted by the singular values $[\sigma_{i1}, ..., \sigma_{id'}]$, forming a hyperellipsoid that reflects the anisotropic density (Fig. 2b). To preserve this anisotropic density, we project the original data points along the local gauges.

Note that both local linear embedding (LLE) (Roweis & Saul, 2000) and local tangent space alignment (LTSA) (Zhang & Zha, 2004) also depict the local manifold structure using tangent space. LLE ensures that each embedding data point is represented as the same linear combination of its neighbors in high-dimensional space, while LTSA implicitly aligns the tangent space to learn the embedding. Our method differs from these approaches by explicitly embedding the tangent spaces with pairwise alignment to construct local gauges in low-dimensional space, enabling the visualization of feature gradients.

### 3.4 Anisotropic projection

In this section, we compute the $d'$-dimensional ($d' \leq d \ll n$) embedding $Y = \{y_1, ..., y_m\}$ under the gauges $V^e$. Each local embedded tangent space $V_i^e$ is weighted by the singular values $\Sigma_i^e = diag(\sigma_{i1}, ..., \sigma_{id'})$. This tangent space locally represents an anisotropic projection in the low-dimensional space.

For the topological embedding $Y = \{y_1, ..., y_m\}$, we borrow the techniques from UMAP (McInnes et al., 2018) with local probability distribution

$$Q_{ij} = (1 + a\|y_i - y_j\|^{2b})^{-1} \tag{12}$$

where $\|y_i - y_j\|$ denotes the Euclidean distance in embedding space, and $a, b$ are shape parameters, and the cross-entropy loss

$$CE(P\|Q) = -\sum_{ij} P_{ij}\log Q_{ij} + (1 - P_{ij})\log(1 - Q_{ij}). \tag{13}$$

Next, we enhance the above loss function by incorporating local anisotropic radius preservation. Consider the low-dimensional data points $y_i$, with the embedding gauges $V_i^e = [v_1^e, ..., v_{d'}^e] \in \mathbf{R}^{d' \times d'}$. The local neighbours $Y_i = \{y_1, ..., y_k\}$ in embedding space are modelled as a hyperellipsoid, with the radius in the $l$-th principal direction $v_l^e$ computed as the weighted average distance[3]:

$$R_{il}^e = \frac{1}{\sum_{j=1}^k Q_{ij}} \sum_{j=1}^k Q_{ij} \|(y_j - y_i)v_l^e\|^2. \tag{14}$$

Recall that the original tangent space $V_i$, with weight $\Sigma_i = diag(\sigma_{i1}, ..., \sigma_{id})$, forms a hyperellipsoid (Fig. 2b) with radius $R_{il}^o = \sigma_{il}^2$ in the $l$-th principal direction. Comparing all directions from $l = 1, 2, ..., d'$, we have

$$Corr(r^o, r^e) = \sum_{l=1}^{d'} Corr(r_l^o, r_l^e) \tag{15}$$

where $Corr(r_l^o, r_l^e)$ is the correlation coefficient of local radii in $l$-th direction between the high- and low-dimensional space, and $r_l^o, r_l^e$ are logarithm of $R_l^o, R_l^e$, respectively.

We combine the correlation coefficient for anisotropic density (Eq. (15)) with the loss function for preserving local pairwise distance (Eq. (13)) to obtain the overall loss function:

$$\mathcal{L} = CE(P\|Q) - \lambda Corr(r^o, r^e) \tag{16}$$

where $\lambda$ controls the relative importance of anisotropic density preservation. We minimize this loss function using SGD to get the $d'$-dimensional embedding coordinates $Y = \{y_1, ..., y_m\}$.

Therefore, the embedding coordinates $Y = \{y_1, ..., y_m\}$ not only preserve the topological structure of local similarity and anisotropic density, but also encapsulate the tangent space embedding $V^e$ which locally demonstrates the feature gradients (Fig. 2d). The pseudo-code for our FeatureMAP algorithm is provided in Appendix A.1.6.

**Training.** Detailed calculation for SGD is provided in Appendix A.1.3. In practice, we use the same (hyper)parameters as UMAP (McInnes et al., 2018), including number of neighbours, number of iterations and the $min\_dist$ parameter. There are two additional parameters: the weight $\lambda \le 0$ for anisotropic density preservation and the fraction $q \in [0, 1]$ of iterations that consider tangent space embedding. For our experiment, we use 15 neighbours, 500 epochs, $q = 0.3$ and $\lambda = 0.5$. Tuning of $\lambda$ is included in Fig. 7.

## 4 EXPERIMENTS

We evaluate FeatureMAP's ability to interpret the embeddings of MNIST digit classification as well as Fashion MNIST and COIL-20 object detection, by using feature gradients. Additionally, we apply FeatureMAP to interpreting MNIST adversarial examples, showing that our method leverages feature importance to explicitly explain misclassification following an adversarial attack. Furthermore, we compare FeatureMAP with state-of-the-art algorithms using both local and global structure preservation metrics. More experiments are included in Appendix A.2.

### 4.1 DATASETS AND EVALUATION METRICS

We perform the evaluation on various datasets including standard MNIST (LeCun, 1998), Fashion MNIST (Xiao et al., 2017), COIL-20 (Nene et al., 1996), Cifar10 (Krizhevsky et al., 2009), single cell RNA-seq (Liu et al., 2021), and MNIST adversarial examples (Goodfellow et al., 2014). We compare with representative methods for dimensionality reduction and visualization, including t-SNE (Van der Maaten & Hinton, 2008), h-NNE (Sarfraz et al., 2022), UMAP (McInnes et al., 2018), triMAP (Amid & Warmuth, 2019), PaCMAP (Wang et al., 2021), densMAP (Narayan et al., 2021) and spaceMAP (Zu & Tao, 2022), PHATE (Moon et al., 2019) in terms of both local and global structure preservation (Espadoto et al., 2019; Wang et al., 2021). Local structure preservation metrics include $k$NN accuracy, trustworthiness (Venna & Kaski, 2006a) and continuity (Venna & Kaski, 2006b). Global structure preservation metrics cover Shepard goodness, normalized stress (Joia et al., 2011) and triplet centroid accuracy (Wang et al., 2021). The details to calculate these metrics are found in (Espadoto et al., 2019; Wang et al., 2021).

---

[3]We use squared distance because of better empirical performance.

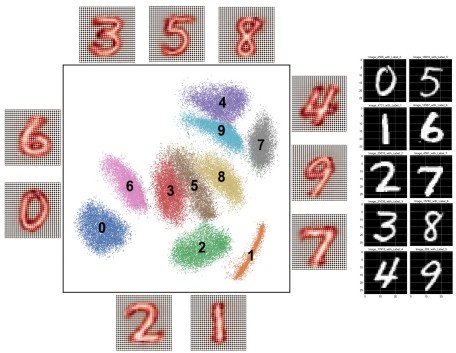

Figure 4: FeatureMAP on MNIST showing feature importance. FeatureMAP embeds the MNIST dataset into 2-dimensional space, forming 10 different clusters (left). From each cluster, digit images are randomly selected to illustrate the feature importance by saliency map, with corresponding original images displayed on the right. Darker red regions indicate greater feature importance.

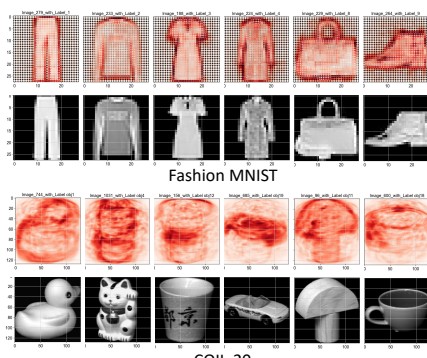

Figure 5: FeatureMAP on Fashion MNIST and COIL-20 showing feature importance. For each dataset, the top row displays the saliency maps indicating feature importance, while the bottom row shows the corresponding original images.

## 4.2 RESULTS

### 4.2.1 FEATUREMAP PRESERVING SOURCE FEATURES

We demonstrate that FeatureMAP enables interpretable dimensionality reduction by feature importance in MNIST digit classification, as well as Fashion MNIST and COIL-20 object detection. Fig. 4 clearly shows distint clusters of different digit groups using FeatureMAP. For each cluster, we randomly select one data point and display its feature importance as saliency map corresponding to the original images. We find that feature importance explicitly highlights the edges of each digit.

Similarly, we apply FeatureMAP to datasets Fashion MNIST and COIL-20 in Fig. 5. In both datasets, the important features succeed in identifying objects across different categories. In the COIL-20 dataset, we also observe that the rotation patterns (lighter red curves) are revealed in the saliency map, reflecting the dataset's generation process, where objects were rotated under a fixed camera (Nene et al., 1996).

We further illustrate the interpretability of FeatureMAP by showing the gradients and feature importance in Fig. 1. Specifically, we annotate the top-10 most important pixel features, which mainly appear at the corner angles of the digit object. This suggests that these features play a dominant role in shaping the local structure of the image. In addition, the edge patterns are observed in the saliency maps of the digit 1.

### 4.2.2 FEATUREMAP INTERPRETING ADVERSARIAL EXAMPLES

We show that FeatureMAP interprets MNIST adversarial examples by explaining the misclassification after adversarial attack. We use Fast Gradient Sign Attack (FGSM) (Goodfellow et al., 2014) to synthesize fake images of MNIST with $\epsilon = 0.3$ and prediction accuracy $0.08$, to fool the classifier LeNet (LeCun et al., 1998). Formally, the adversarial example $x' \in \mathbb{R}^n$ satisfies $\|x' - x\| \leq \epsilon$, where $x$ is the original image, and the predicted label $p(x') \neq p(x)$. The results are included in Fig. 6 and Supplementary Fig. 8.

We plot the adversarial examples by FeatureMAP in Fig. 6. The top-left part displays both the original and adversarial labels. Notably, the clustering structure of the adversarial examples closely resembles the original clusters shown in Fig. 4. This similarity arises because the adversarial examples fall within the $\epsilon$-neighborhood of the original data points ($\epsilon = 0.3$), which makes the adversarial examples retain similar local topological structure with the original examples.

FGSM successfully fools the classifier, with $43\%$ of the data misclassified as digit 8. We randomly select five images from the cluster originally labeled as digit 1 and display the corresponding adver-

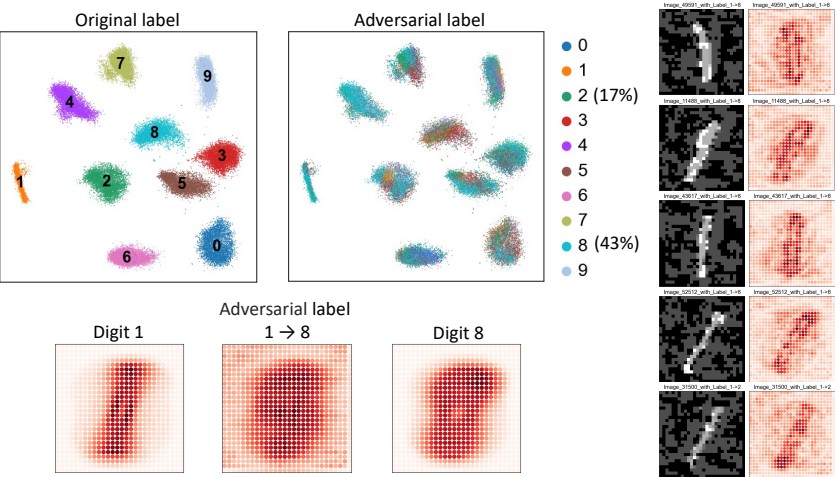

Figure 6: FeatureMAP on MNIST adversarial examples. FeatureMAP embeds MNIST adversarial examples to 2-dimensional space, displaying both the original and adversarial labels (top-left). On the right, five randomly selected adversarial examples from original digit 1 group are shown. The bottom-left visualizes the average feature importance of adversarial examples misclassified from 1 to 8, alongside the corresponding original digits 1 and 8 before the attack. The saliency map of the adversarial examples closely resembles that of digit 8.

sarial images with feature importance on the right of Fig. 6. Although the adversarial images are visually recognizable as digit 1, the saliency map highlights additional important features beyond the typical edge features of digit 1.

To interpret the additional important features, we compute the average feature importance for adversarial examples misclassified from digit 1 to 8, and compare it with the original average feature importance of digits 1 and 8 (bottom-left in Fig. 6), respectively. The results show that the important feature patterns in the adversarial examples more closely resemble those of the original digit 8 than digit 1, providing a clear explanation for the misclassification after the attack.

### 4.3 QUANTITATIVE COMPARISON WITH STATE-OF-THE-ART

We compare FeatureMAP with other representative dimensionality reduction methods. The two-dimensional visualization plots are provided in Fig. 13, and the quantitative comparison results are summarized in Tab. 1. FeatureMAP demonstrates comparable performance on both local and global metrics. Additionally, FeatureMAP outperforms UMAP in terms of density preservation (Figs. 9, 10 and 12). We include the running time comparison in Fig. 14.

## 5 DISCUSSION

We propose FeatureMAP, an interpretable nonlinear dimensionality reduction method that employs manifold gradients to explain the low-dimensional embedding space. Specifically, we embed tangent spaces to locally construct gauges that display feature gradients, thereby explaining the reduced-dimensional results through feature directions and importance. FeatureMAP also maintains anisotropic density in two-dimensional plots, which further enhances the interpretability of the visualization. Our method of preserving features through tangent space embedding provides a plug-in module for manifold learning.

Experiments on digit classification, object detection, and MNIST adversarial examples demonstrate that FeatureMAP produces interpretable dimensionality reduction results, which facilitate the explanation of classification outcomes and feature detection. In future work, we plan to extend our feature-preserving paradigm by applying tangent space embedding to other nonlinear dimensionality reduction methods to enhance interpretability. Additionally, we will apply FeatureMAP to real-world image datasets such as CIFAR-10 and ImageNet, as well as to biological gene expression data, to strengthen interpretation in classification and feature detection tasks. Furthermore, we intend to explore complex data by learning sparse, high-order features through sparse manifold transforms (Chen et al., 2018; 2022).

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

# A  APPENDIX

## A.1  SOME DETAILS ON FEATUREMAP ALGORITHMS

### A.1.1  FUNCTION GRADIENT OVER A MANIFOLD

Given a smooth function $f$ on a Riemannian manifold $(\mathcal{M}, g)$, the gradient of $f$, denoted $\nabla f$, is the unique vector field satisfying: $g(\nabla f, X) = df(X)$, for all vector fields $X$ on $\mathcal{M}$, where $g$ is the Riemannian metric, $df$ is the differential of $f$, and $g(\nabla f, X)$ denotes the inner product between $\nabla f$ and $X$ at each point.

We now illustrate how to estimate the gradient with respect to a single feature $f_j$ from discrete data, where the function of interest is $f_j(x) = x_j$. Clearly, the gradient vector of a smooth function resides in the tangent space at each point on the manifold.

To approximate the tangent space at one data point $x_i$, we perform local PCA (see Equation 2), obtaining the basis vectors $\{v_1, v_2, ..., v_d\}$. Under this orthonormal basis, the Riemannian metric simplifies to the identity matrix, $g = I$.

Next, we compute the gradient of function $f_j$ as $df_j(x) = e_j$, where $e_j$ is the $j$-th standard basis vector. We then project this gradient to PCA basis by calculating the coefficients $c_i = \langle df_j, v_i \rangle = v_{ij}$. Thus, the gradient in PCA basis is given by

$$\nabla f_j = \sum_{i=1}^{d} c_i v_i = \sum_{i=1}^{d} v_{ij} v_i \tag{17}$$

This result indicates the direction of steepest increase in terms of PCA basis.

Finally, the magnitude of the gradient is

$$||\nabla f_j|| = (\sum_{i=1}^{d} v_{ij}^2)^{\frac{1}{2}} \tag{18}$$

This value represents the rate of increase of $f_j$ in the direction of $\nabla f_j$.

### A.1.2 LOCAL RADIUS PRESERVATION

We augment loss function of distance preservation by incorporating local anisotropic radius preservation. Consider the low-dimensional data points $y_i$ with the embedding gauges $V_i^e = [v_1^e, ..., v_{d'}^e] \in \mathbf{R}^{d' \times d'}$. We center its $k$ nearest neighbours as $Y_i = [y_1 - y_i, ..., y_k - y_i]^T \in \mathbb{R}^{k \times d'}$, and project $Y_i$ to its tangent space $V_i^e$ as $\hat{Y}_i = Y_i V_i^e$. For one of $y_i$'s neighbours, say $y_j$, the distance $||y_i - y_j||$ is invariant under the rotation $V_i^e$, which makes $\hat{Y}_i$ fit Eq. (12).

Recall that the original tangent space $V_i$ with weight $\Sigma_i = diag(\sigma_{i1}, ..., \sigma_{id})$ forms a hyperellipsoid (Fig. 2b) with radius $R_{il}^o = \sigma_{il}^2, l = 1, ..., d$[4]. The local neighbours $Y_i$ in embedding space is also modelled as a hyperellipsoid, with the radius in direction $v_{il}^e$ ($l = 1, ..., d'$) as

$$R_{il}^e = \frac{1}{\sum_j Q_{ij}} \sum_j Q_{ij} ||(y_j - y_i) v_{il}^e||^2, \tag{19}$$

To preserve the local radii in the low-dimensional embedding space, we project the data points to ensure that the volume of the local point cloud in the low-dimensional space matches the original hyperellipsoid in high-dimensional space. Formally, we set the local hyperellipsoid volume of original and embedding space by $c(d) \prod_{l=1}^{d} R_{il}^o = c(d') \prod_{l=1}^{d'} R_{il}^e$, where $c$ denotes a constant related to dimensions. We rewrite this equation by logarithm and get

$$r_{i1}^o + ... + r_{id'}^o + c = r_{i1}^e + ... + r_{id'}^e \tag{20}$$

where $r_{il}^o = \log R_{il}^o$ and $r_{il}^e = \log R_{il}^e$. It is sufficient to set $r_{il}^e = \beta r_{il}^o + \alpha$ ($l = 1, ..., d'$) to achieve the above equation, where the problem is modified to a affine relationship between the local radii. In the $l$-th principal direction, we measure the goodness of fitting this relationship by correlation coefficient

$$Corr(r_l^o, r_l^e) = \frac{Cov(r_l^o, r_l^e)}{(Var(r_l^o) Var(r_l^e))^{1/2}} \tag{21}$$

where $r_l^o = [r_{1l}^o, ..., r_{ml}^o]$ and $r_l^e = [r_{1l}^e, ..., r_{ml}^e]$. For all directions, we have

$$Corr(r^o, r^e) = \sum_{l=1}^{d'} Corr(r_l^o, r_l^e) \tag{22}$$

where $Corr(r^o, r^e)$ indicates the agreement of local radii between the high-dimensional and low-dimensional space.

---

[4] We use squared distance (variance) because of better empirical performance.

### A.1.3 STOCHASTIC GRADIENT DESCENT (SGD) FOR ANISOTROPIC PROJECTION

We use anisotropic projection to embed the data points into low-dimensional space. To apply SGD to optimizing the embedding coordinates, we calculate the derivative of the loss function

$$\mathcal{L} = CE(P||Q) - \lambda Corr(r^o, r^e). \tag{23}$$

The core of the anisotropic projection lies in optimizing the Pearson correlation between the local radius in original and embedding space. We first calculate the gradient of this correlation regarding the embedding coordinates for optimization. We rewrite the correlation as follows:

$$Corr(r^o, r^e) = \sum_{l=1}^{d'} \frac{Cov(r_l^o, r_l^e)}{(Var(r_l^o) Var(r_l^e))^{1/2}} \tag{24}$$

where $r_l^o = \{r_{il}^o\}_{i=1}^m = \{log R_{il}^o\}_{i=1}^m$ and $r_l^e = \{r_{il}^e\}_{i=1}^m = \{log R_{il}^e\}_{i=1}^m$ in the $l$-th principal direction of original and embedding spaces respectively. We set $z_{ij} = y_j - y_i$ and rewrite $r_{il}^e$ as

$$\begin{aligned} r_{il}^e = log R_{il}^e &= log \frac{1}{\sum_j Q_{ij}} \sum_j Q_{ij} \|(y_j - y_i)v_{il}^e\|^2 \\ &= log \frac{1}{\sum_j Q_{ij}} \sum_j Q_{ij} \|z_{ij} v_{il}^e\|^2, \end{aligned} \tag{25}$$

and

$$Q_{ij} = \frac{1}{1 + a(y_j - y_i)^{2b}} = [1 + a(z_{ij}^T z_{ij})^b]^{-1} \tag{26}$$

Let

$$\rho_{e,o}^{(l)} = \frac{Cov(r_l^o, r_l^e)}{(Var(r_l^o) Var(r_l^e))^{1/2}}, \tag{27}$$

be the correlation in the $l$-th direction. The derivative of $\rho_{e,o}^{(l)}$ with respect to $z_{ij}$ is

$$\frac{\partial \rho_{e,o}^{(l)}}{\partial z_{ij}} = Var(r_l^o)^{-1/2} \left[ \frac{\partial Cov(r_l^o, r_l^e)}{\partial z_{ij}} Var(r_l^e)^{-1/2} - \frac{1}{2} Cov(r_l^o, r_l^e) Var(r_l^e)^{-3/2} \frac{\partial Var(r_l^e)}{\partial z_{ij}} \right] \tag{28}$$

Therefore, the gradient of the correlation in Eq. (15) in the $l$-th principal direction regarding the embedding coordinates $y_i$ is

$$\nabla_{y_i} Corr^{(l)}(r^o, r^e) = \sum_{j \neq i} \frac{\partial \rho_{e,o}^{(l)}}{\partial z_{ij}} \frac{\partial z_{ij}}{\partial y_i} \tag{29}$$

We further compute the derivative of the variance and covariance in Eq. (28). To simplify the notation, we set $\mu_l^e = \mathbb{E}[r_l^e]$ and center the original local radius as $r_{il}^o := log R_{il}^o - m^{-1} \sum_{i=1}^m log R_{il}^o$. The gradient in Eq. (28) ($l$ omitted) is calculated as

$$\begin{aligned} \frac{\partial \rho_{e,o}^{(l)}}{\partial z_{ij}} = Var(r^o)^{-\frac{1}{2}} \Big\{ &\frac{1}{m-1}(r_i^o \frac{\partial r_i^e}{\partial z_{ij}} + r_j^o \frac{\partial r_j^e}{\partial z_{ij}}) Var(r_l^e)^{-\frac{1}{2}} \\ &- \frac{1}{m-1} Cov(r^o, r^e) Var(r^e)^{-\frac{3}{2}} [(r_i^e - \mu^e) \frac{\partial r_i^e}{\partial z_{ij}} \\ &+ (r_j^e - \mu^e) \frac{\partial r_j^e}{\partial z_{ij}}] \Big\}, \end{aligned} \tag{30}$$

and

$$\frac{\partial r_i^e}{\partial z_{ij}} = \frac{\tilde{Q}_{ij}^2 W_i}{R_i^e} [(1 + a(z_{ij}^T z_{ij})^b - 2ab(z_{ij}^T z_{ij})^{b-1} z_{ij} z_{ij}^T) 2v_i v_i^T z_{ij}] + \tilde{Q}_{ij}^2 W_i * 2ab(z_{ij}^T z_{ij})^{b-1} z_{ij} \tag{31}$$

where $W_i = \sum_{s=1}(1 + a(z_{is}^T z_{is})^b)^{-1}$ and $\tilde{Q}_{ij} = W_i^{-1}(1 + a(z_{ij}^T z_{ij})^b)^{-1}$.

Next, we compute the gradient of the cross-entropy $CE(P||Q)$ (similar to UMAP (McInnes et al., 2018)). The cross-entropy loss function is

$$\mathcal{L}_{ce} = CE(P||Q) = -\sum_{ij} P_{ij} log Q_{ij} + (1 - P_{ij}) log(1 - Q_{ij}) \tag{32}$$

and its gradient with respect to $z_{ij}$ is

$$\frac{\partial \mathcal{L}_{ce}}{\partial z_{ij}} = -[P_{ij} \frac{\partial log Q_{ij}}{\partial z_{ij}} + (1 - P_{ij}) \frac{\partial log(1 - Q_{ij})}{\partial z_{ij}}] \tag{33}$$

where $\frac{\partial log Q_{ij}}{\partial z_{ij}} = \frac{-2ab(z_{ij}^T z_{ij})^{b-1} z_{ij}}{1 + a(z_{ij}^T z_{ij})^b}$ and $\frac{\partial log(1-Q_{ij})}{\partial z_{ij}} = \frac{1}{z_{ij}^T z_{ij}} \frac{2bz_{ij}}{1+(z_{ij}^T z_{ij})^b}$.

The attractive term $P_{ij} log Q_{ij}$ is optimized by randomly drawing an edge $(i, j)$ by distribution $P$, which means that the edge $(i, j)$ is selected with probability $\frac{P_{ij}}{\sum_{i \neq j} P_{ij}}$, followed by calculating the gradient $\frac{\partial log Q_{ij}}{\partial z_{ij}}$. The repulsive term is estimated by uniformly at random choosing a set of points $S$ adjacent to the given point $x_i$ and computing $\frac{1}{|S|} \sum_{l \in S} \frac{\partial log(1 - Q_{il})}{\partial z_{il}}$.

Combine the gradient of cross-entropy in Eq. (33) and local radius correlation in Eq. (30). The gradient for an edge $(i, j)$ at each iteration of the SGD in the $l$-th principal direction is

$$\nabla_{y_i} \mathcal{L}|_{(i,j)} = -[\frac{\partial log Q_{ij}}{\partial z_{ij}} - \frac{1}{|S|} \sum_{l \in S} \frac{\partial log(1 - Q_{il})}{\partial z_{il}} + \lambda \frac{Z}{mP_{ij}} \frac{\partial \rho_{e,o}^{(l)}}{\partial z_{ij}}] \frac{\partial z_{ij}}{\partial y_i}, \tag{34}$$

where $Z = \sum_{(i,j)} P_{ij}$ and $\frac{mP_{ij}}{Z}$ is the normalization factor considering the edge is chosen with probability $\frac{P_{ij}}{Z}$.

In addition, we compute the gradient in Eq. (30) at the start of each epoch. We achieve this by calculating $W_i = \sum_{s=1} Q_{is}$, the local radius $r_{il}^e$, the variance and covariance terms in the beginning of each epoch, regarding them as fixed for all the edges to be updated during that epoch.

A.1.4 STOCHASTIC GRADIENT DESCENT (SGD) FOR TANGENT SPACE EMBEDDING

The key of tangent space embedding is to preserve the alignment between tangent spaces. We depict the alignment by the general angle between two tangent spaces, which induces the cosine distance of tangent spaces. We further model the probability distribution of cosine distance in original and embedding space respectively, and minimize the difference of distribution by KL divergence to maintain the alignment in embedding space.

In practice, we borrow the framework of t-SNE or UMAP to compute the embedding tangent space. We flat the tensor of tangent spaces in original space (i.e., $V_i$ and $V_j$) as $v_i$ and $v_j$ and feed them to t-SNE as input. We set the parameter *distance* as *cosine*. The output is normalized to get the embedding tangent space.

Specifically, we apply SGD to the loss function of Eq. (11) to update the tangent space embedding. Let $\mathcal{L}_{KL} = \mathbf{S}||\mathbf{T}) = \sum_{ij} \mathbf{S}_{ij}(\log \mathbf{S}_{ij} - \log \mathbf{T}_{ij})$. For the clarity of computation, we set $t_{ij} = d_{ij}^2$ and calculate the derivative of $\mathcal{L}_{KL}$ with respect to $t_{ij}$ as

$$\frac{\partial \mathcal{L}_{KL}}{\partial t_{ij}} = -S_{ij} \frac{\partial log T_{ij}}{\partial t_{ij}} \tag{35}$$

where $\frac{\partial log T_{ij}}{\partial t_{ij}} = \frac{-abt_{ij}^{b-1}}{1 + at_{ij}^b}$. The derivative of $t_{ij}$ with respect to $v_i$ is $\frac{\partial t_{ij}}{\partial \mathbf{v}_i} = \frac{-\|\mathbf{v}_i\|\|\mathbf{v}_j\| + \frac{\mathbf{v}_i}{\|\mathbf{v}_i\|}(\mathbf{v}_i^\top \mathbf{v}_j)}{\|\mathbf{v}_i\|^2 \|\mathbf{v}_j\|}$.

Thus, the gradient for an edge $(i, j)$ at each iteration of SGD is

$$\nabla_{v_i} \mathcal{L}|_{(i,j)} = -[\frac{\partial log Q_{ij}}{\partial t_{ij}}] \frac{\partial t_{ij}}{\partial v_i}. \tag{36}$$

### A.1.5 HYPERPARAMETER TUNING

In this section, we discuss how to set the hyperparameters including $n\_neighbors$, $min\_dist$ and $\lambda$, and their impact on FeatureMAP's performance.

The $n\_neighbors$ parameter determines how FeatureMAP balances local and global structures by defining the neighborhood size for learning the manifold. Smaller values capture local details but may miss the overall structure, while larger values prioritize the broader structure at the expense of fine details. Following UMAP and t-SNE conventions, where $n\_neighbors$ is typically 15 or 30, we set $n\_neighbors = 15$ for FeatureMAP.

The $min\_dist$ parameter controls point spacing in FeatureMAP embeddings. Lower values create compact clusters, useful for clustering and fine details. Higher values preserve broader topological structures by spreading points further apart.

We demonstrate the tuning of the hyperparameter $\lambda$ in the loss function of FeatureMAP in Fig. 7. The parameter $\lambda = 0$ corresponds to UMAP. With the increasing of $\lambda$, FeatureMAP performs better in preserving the local anisotropic density with the local radius correlation getting larger, while the clusters in the two-dimensional plot are not clearly separable when $\lambda$ is large. Consider the trade-off between the local density preservation and clusters visualization, we set the parameter $\lambda$ as $0.5$.

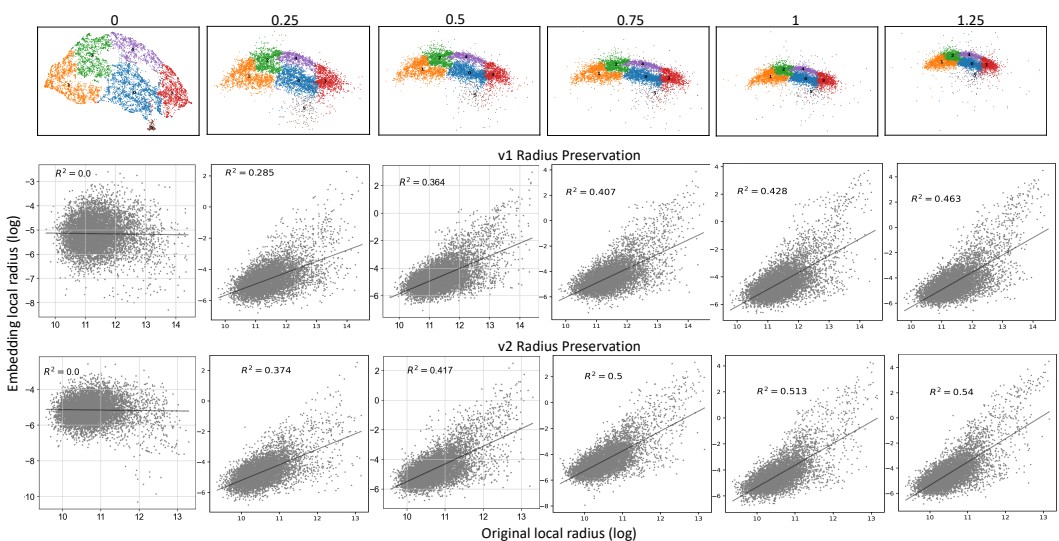

Figure 7: The tuning of hyperparameter $\lambda$. The hyperparameter $\lambda$ in FeatureMAP is set from $0$ to $1.25$ with two-dimensional plot and local radius correlation.

### A.1.6 ALGORITHM PSEUDOCODE

Here are the pseudocodes for our FeatureMAP algorithm. The main algorithm is divided into seven steps, each detailed in a separate pseudocode.

---

**Algorithm 1** FeatureMAP Algorithm

---

1: **Input:** Data points $X = \{x_1, \ldots, x_m\} \subset \mathbb{R}^n$, number of neighbors $k$, target dimensionality $d'$
2: **Output:** Low-dimensional embedding $Y = \{y_1, \ldots, y_m\} \subset \mathbb{R}^{d'}$, gradients, and feature importance
3:
4: **1. Construct the k-nearest neighbor (kNN) graph (Algorithm 2)**
5: **2. Compute tangent spaces at each data point (Algorithm 3)**
6: **3. Align tangent spaces (Algorithm 4)**
7: **4. Embed tangent spaces to low-dimensional space (Algorithm 5)**
8: **5. Project data points anisotropically (Algorithm 6)**
9: **6. Compute gradients and feature importance (Algorithm 7)**
10: **7. Return $Y$, gradients, and feature importance**

---

**Algorithm 2** Construct kNN Graph

---

1: **Input:** Data points $X = \{x_1, \ldots, x_m\}$, number of neighbors $k$
2: **Output:** kNN graph $G(X, E)$ with edge weights $P_{ij}$
3: **for** each pair of data points $(x_i, x_j)$ **do**
4:     Find the k-nearest neighbors for $x_i$
5:     Connect $x_i$ to its k-nearest neighbors to form graph $G(X, E)$
6:     Compute edge weight $P_{ij}$ based on geodesic distance (Eq. (1))
7: **end for**
8: **Return:** $G(X, E)$ and $P_{ij}$

---

**Algorithm 3** Compute Tangent Spaces

---

1: **Input:** Data points $X$, kNN graph $G(X, E)$
2: **Output:** Tangent spaces $T_{x_i}$ for each $x_i$
3: **for** each data point $x_i$ **do**
4:     Select its k-nearest neighbors
5:     Compute weighted singular value decomposition (SVD) of the neighborhood data matrix
6:     Extract the principal components (right singular vectors) as tangent space $T_{x_i}$ (Eq. (2))
7: **end for**
8: **Return:** $T_{x_i}$ for all $x_i$

---

---

**Algorithm 4** Align Tangent Spaces

---

1: **Input:** Tangent spaces $T_{x_i}$, kNN graph $G(X, E)$
2: **Output:** Alignment matrices $O_{ij}$
3: **for** each pair of neighboring data points $(x_i, x_j)$ **do**
4:     Compute the alignment matrix $O_{ij}$ using orthonormal bases of $T_{x_i}$ and $T_{x_j}$ (Eq. (5))
5:     Ensure alignment by preserving cosine distance between tangent spaces (Eqs. (7) and (8))
6: **end for**
7: **Return:** $O_{ij}$ for all pairs

---

**Algorithm 5** Embed Tangent Spaces

---

1: **Input:** Tangent spaces $T_{x_i}$, alignment matrices $O_{ij}$
2: **Output:** Embedded tangent spaces in low-dimensional space $V_e$
3: **for** each tangent space $T_{x_i}$ **do**
4:     Embed tangent space $T_{x_i}$ into low-dimensional space $V_e$, preserving cosine similarity (Eqs. (9) and (10))
5:     Maximize agreement between original and embedded tangent spaces using Kullback-Leibler divergence (Eq. (11))
6:     Optimize embedding using stochastic gradient descent (SGD)
7: **end for**
8: **Return:** $V_e$ for all tangent spaces

---

**Algorithm 6** Anisotropic Projection

---

1: **Input:** Data points $X$, embedded tangent spaces $V_e$
2: **Output:** Low-dimensional embedding $Y$
3: **for** each data point $x_i$ **do**
4:     Project data point along its embedded tangent space to low-dimensional space (Eq. (12))
5:     Preserve pairwise distances by minimizing cross-entropy loss (Eq. (13))
6:     Preserve local neighborhood volume using hyperellipsoid constraints (Eqs. (14) to (16))
7: **end for**
8: **Return:** $Y$

---

**Algorithm 7** Compute Gradients and Feature Importance

---

1: **Input:** Data points $X$, embedded tangent spaces $V_e$
2: **Output:** Gradients and feature importance for each data point
3: **for** each data point $x_i$ **do**
4:     **for** each feature $f_j$ **do**
5:         Compute gradient of $f_j$ in tangent space (Eq. (3))
6:         Use gradient magnitudes to compute feature importance (Eq. (4))
7:         Visualize feature importance with saliency maps and quiver plots
8:     **end for**
9: **end for**
10: **Return:** Gradients and feature importance

---

### A.1.7 COMPARATIVE SUMMARY OF FEATUREMAP'S ADVANTAGES

| Method | Strengths | Limitations | FeatureMAP Advantage |
|---|---|---|---|
| **UMAP** | Fast, scalable, preserves neighborhood connectivity | Lacks explicit feature preservation mechanisms | Embeds both tangent spaces and kNN graphs for feature visualization |
| **densMAP** | Preserves density variations | Does not account for feature relationships explicitly | Explicitly incorporates feature-level geometry |
| **PHATE** | Captures temporal and trajectory structure | Computationally expensive, lacks feature interpretability | Combines scalability with feature interpretability |
| **t-SNE** | Excellent at preserving local structures | Computationally expensive, struggles with global structure, lacks feature interpretability | Balances local and global structure while preserving feature relationships |
| **SpaceMAP** | Addresses "crowding problem," robust global structure preservation with Equivalent Extended Distance (EED), adapts to diverse manifolds with hierarchical approximation. | Limited emphasis on feature-level preservation, primarily focuses on global geometry and density. | Preserves both local and global structures, explicitly embeds features for interpretability, enabling feature visualization. |

### A.2 MORE RESULTS ON EXPERIMENTS

#### A.2.1 FEATUREMAP INTERPRETING ADVERSARIAL EXAMPLES

We illustrate how the digits 1 are misclassified to the other labels in Fig. 8. The saliency map illustrates the average feature importance. The feature importance pattern in the middle is more similar to the right than the left, indicating that the adversarial attack FGSM alters the feature importance and fools the LeNet classifier.

#### A.2.2 FEATUREMAP PRESERVING DENSITY

We illustrate that FeatureMAP maintains original data density in two-dimensional plot. We apply FeatureMAP to the digit 1 group of MNIST dataset and plot the correlation of local radius between embedding and original space. FeatureMAP presents larger local radius correlation than UMAP (in Supplementary Fig. 10), indicating that FeatureMAP performs better in local density preservation. Particularly, we cluster the dataset and find that the subgroup (in brown colour) is significantly more sparse in FeatureMAP than UMAP in Fig. 9. We claim that FeatureMAP correctly reveals this sparse pattern because FeatureMAP shows positive correlation of local radius between embedding and original space. We further demonstrate the digit images from this subgroup in Supplementary Fig. 12, which exhibit diverse handwritten patterns and agree with the sparse pattern in FeatureMAP.

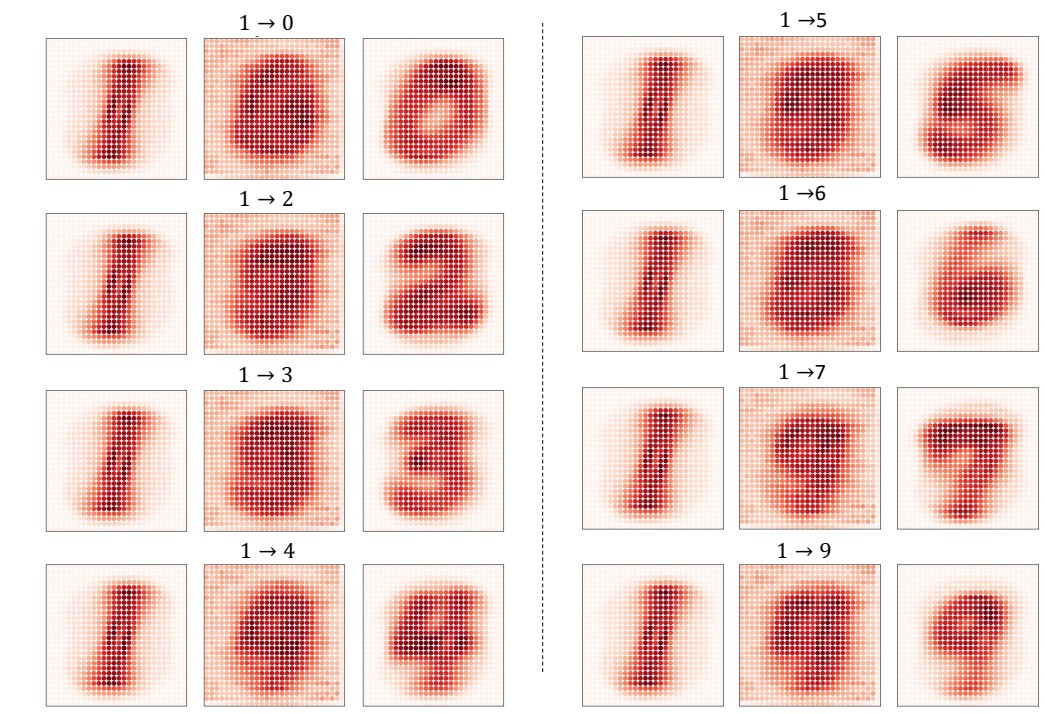

Figure 8: Average feature importance of the adversarial examples and the corresponding original data. For the original digits 1, we list all the corresponding adversarial examples with average feature importance in the middle. The left is the original digit 1 and the right is the original digit with the label same as the corresponding adversarial label.

.

FeatureMAP presents larger local radius correlation than UMAP in Fig. 10. We cluster the data (digit 1 in MNIST) on the right of Fig. 10 and find that the subgroup 5 (in brown colour) is significantly more sparse in FeatureMAP than UMAP.

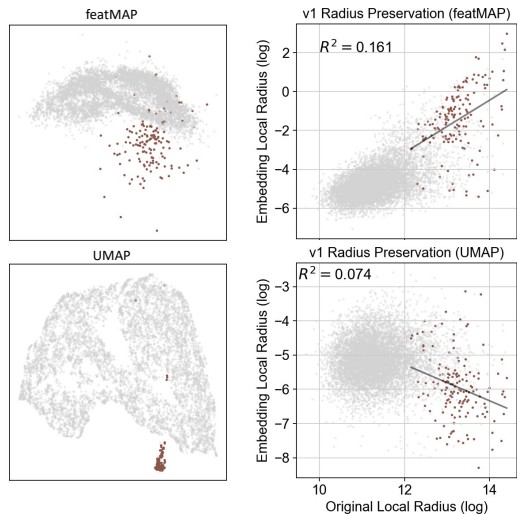

Figure 9: FeatureMAP on MNIST (digit 1) showing density preservation. For one subgroup (colour in brown), FeatureMAP (top) preserves the local radius against UMAP (bottom) by positive correlation between embedding and original local radius; FeatureMAP correctly illustrates the sparse pattern compared to UMAP.

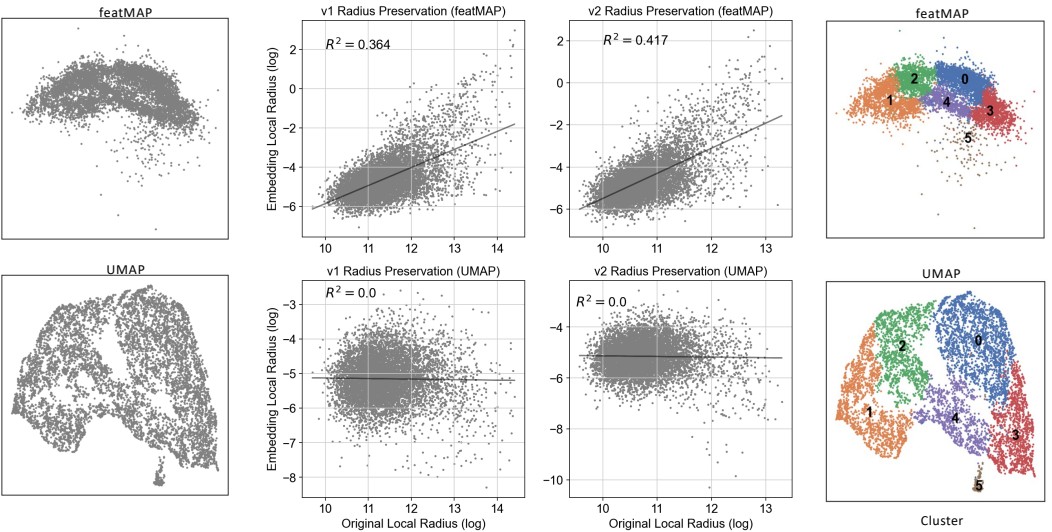

Figure 10: FeatureMAP preserving the original density. FeatureMAP and UMAP are applied to digit 1 group of MNIST (left). The middle is the scatter plot of embedding local radius against original local radius, along with the straight line of linear regression showing the correlation of local radius between embedding and original space. The right illustrates the clusters on both FeatureMAP and UMAP.

### A.2.3    FEATURE IMPORTANCE ON MNIST DATA

We randomly select 20 samples from the MNIST dataset and visualize both the original images and their corresponding saliency maps. The saliency maps effectively highlight the feature importance, illustrating their contribution to the classification.

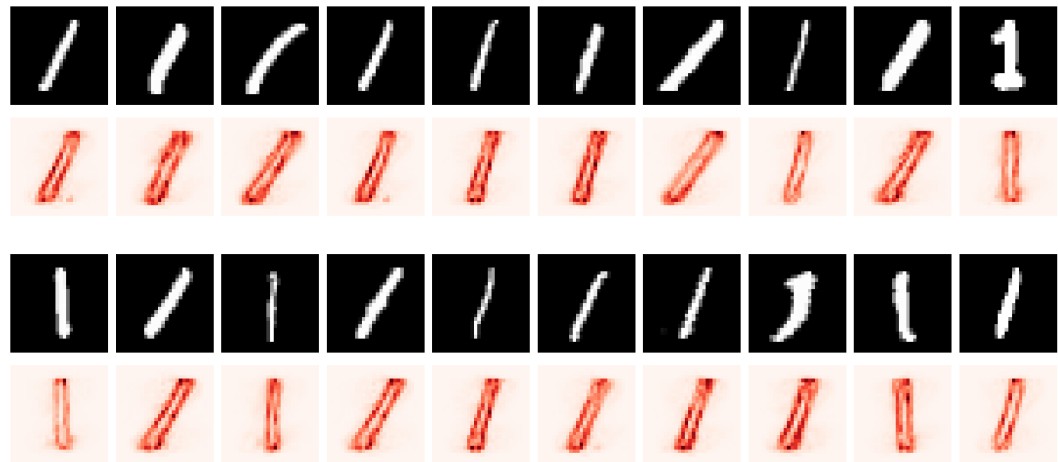

Figure 11: Twenty data points are randomly selected from the MNIST dataset, with their corresponding saliency maps highlighting feature importance.

We further demonstrate the digit images from this subgroup in Fig. 12, which exhibit diverse handwritten patterns and agree with the sparse pattern in FeatureMAP.

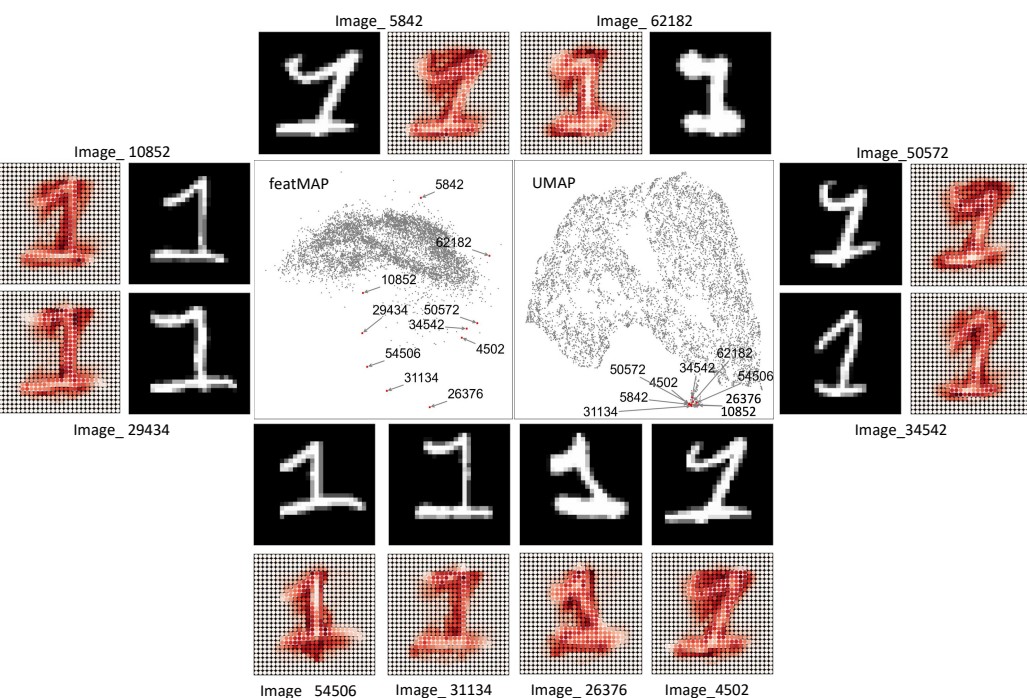

Figure 12: Handwritten digit images from the sparse subgroup of FeatureMAP. 10 data points are randomly selected from the subgroup 5 in Fig. 10 with the corresponding digit images.

### A.2.4 COMPARING FEATUREMAP WITH OTHER STATE-OF-THE-ART

We employ state-of-the-art nonlinear dimensionality reduction methods on benchmark datasets to generate the visualizations shown in Fig. 13. FeatureMAP demonstrates comparable density preservation to densMAP.

We also perform a quantitative comparison of FeatureMAP with other methods in terms of local and global accuracy preservation, as shown in Tab. 1. Overall, FeatureMAP illustrates performance comparable to other methods. On the MNIST and Fashion MNIST datasets, FeatureMAP outperforms the others in local metrics such as continuity and kNN accuracy, notably surpassing densMAP, spaceMAP, and PHATE. For global metrics, FeatureMAP consistently ranks either first or second across measures such as Shepard goodness, normalized stress, and centroid triplet accuracy.

Furthermore, we assess the efficiency of FeatureMAP in comparison with other methods Fig. 14. The running time of FeatureMAP is comparable to that of densMAP and TriMAP, as all three require additional computations beyond kNN graph embedding. Specifically, densMAP involves local density calculations, while TriMAP calculates distances within triplets. While FeatureMAP requires more time than UMAP (e.g., 30s for UMAP vs. 150s for FeatureMAP on 50,000 samples), this is reasonable given its extended functionality. Unlike UMAP, which focuses solely on kNN graph embedding, FeatureMAP also embeds tangent spaces and facilitates feature visualization, justifying the additional computational cost. Moreover, PHATE performs the worst when the data size exceeds 50,000. This is attributed to its reliance on the embedding technique of Multi-dimensional Scaling (MDS), which requires computing all pairwise distances, resulting in significantly increased computational overhead for larger datasets.

**Optimization Strategies to Improve Efficiency**

To enhance efficiency for large datasets, we consider applying the following strategies in our future work:

- Approximate Nearest Neighbor Search: Utilize efficient libraries like Annoy or FAISS to accelerate kNN graph construction without significant accuracy loss.

- Dimensionality Reduction Before Embedding: Apply initial dimensionality reduction (e.g., PCA) to reduce the feature space size, lowering the computational cost for tangent space embedding.

- Parallel Computing: Implement parallelization for distance calculations and graph-based operations to leverage multi-core CPUs or GPUs.

- Batch Processing: Divide the dataset into batches to reduce memory usage during embedding, especially when dealing with extremely large datasets.

- Adaptive Neighborhood Size: Dynamically adjust the $n\_neighbors$ parameter based on the dataset size and structure to balance runtime and embedding quality.

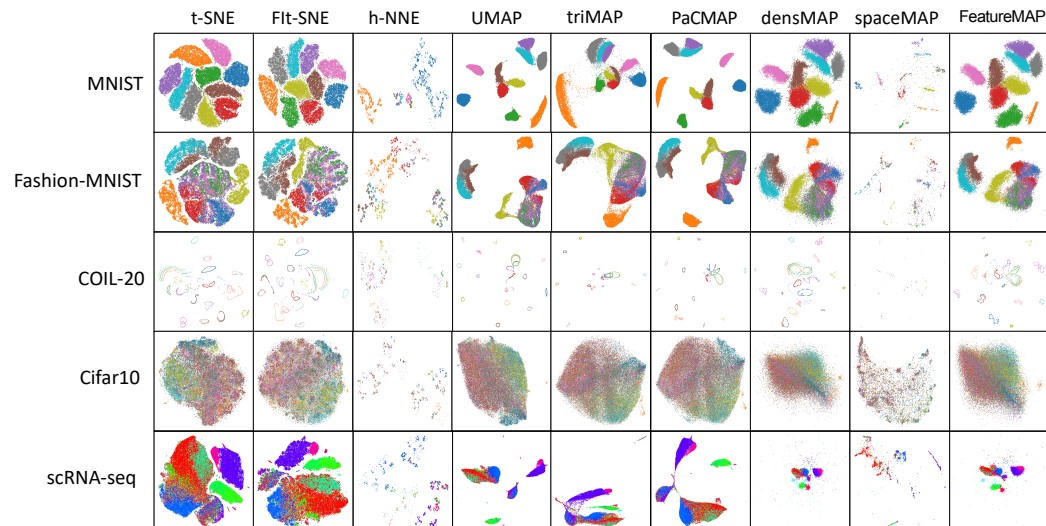

Figure 13: Visualization of multiple datasets by the state-of-the-art nonlinear dimensionality reduction methods.

Table 1: Quantitative comparison. $M_t$, $M_c$ and $M_k$ represent the local metrics: trustworthiness, continuity and $k$NN accuracy, respectively. Similarly, $M_s$, $M_\sigma$ and $M_{ct}$ denote the global metrics: Shepard goodness, normalized stress and centroid triplet accuracy, respectively.

| Experiments | | t-SNE | h-NNE | UMAP | triMAP | PaCMAP | densMAP | spaceMAP | PHATE | FeatureMAP |
|---|---|---|---|---|---|---|---|---|---|---|
| MNIST | $M_t$ | **0.98** | 0.96 | 0.96 | 0.95 | 0.95 | 0.93 | 0.95 | 0.83 | 0.95 |
| | $M_c$ | 0.97 | 0.93 | 0.97 | 0.96 | 0.96 | 0.97 | 0.96 | 0.88 | **0.98** |
| | $M_k$ | 0.97 | 0.96 | 0.97 | 0.96 | 0.97 | 0.96 | 0.97 | 0.76 | **0.97** |
| Fashion MNIST | $M_t$ | **0.99** | 0.95 | 0.98 | 0.97 | 0.97 | 0.96 | 0.97 | 0.83 | 0.96 |
| | $M_c$ | 0.99 | 0.95 | **0.99** | **0.99** | 0.98 | 0.98 | 0.98 | 0.86 | **0.99** |
| | $M_k$ | 0.82 | **0.84** | 0.79 | 0.76 | 0.77 | 0.76 | 0.80 | 0.62 | **0.84** |
| COIL-20 | $M_t$ | **1.00** | 0.98 | 0.99 | 0.99 | 0.99 | 0.98 | 0.98 | 0.95 | 0.99 |
| | $M_c$ | 0.99 | 0.97 | **1.00** | 0.99 | 0.99 | 1.00 | 0.99 | 0.97 | 0.99 |
| | $M_k$ | **0.95** | 0.89 | 0.87 | 0.82 | 0.84 | 0.92 | 0.86 | 0.80 | 0.94 |
| Cifar10 | $M_t$ | **0.93** | 0.83 | 0.84 | 0.85 | 0.84 | 0.80 | 0.90 | 0.83 | 0.92 |
| | $M_c$ | 0.92 | 0.76 | **0.94** | 0.94 | 0.94 | 0.93 | 0.92 | 0.91 | **0.94** |
| | $M_k$ | 0.82 | **0.84** | 0.79 | 0.76 | 0.77 | 0.77 | 0.31 | 0.75 | 0.83 |
| RNA-seq | $M_t$ | **0.99** | 0.96 | 0.96 | 0.95 | 0.95 | 0.93 | 0.98 | 0.82 | 0.98 |
| | $M_c$ | 0.97 | 0.94 | 0.97 | 0.96 | 0.96 | **0.97** | 0.97 | 0.90 | 0.97 |
| | $M_k$ | 0.66 | 0.69 | 0.64 | 0.63 | 0.62 | 0.62 | 0.67 | 0.76 | **0.70** |

(a) Local metrics

| Experiments | | t-SNE | h-NNE | UMAP | triMAP | PaCMAP | densMAP | spaceMAP | PHATE | FeatureMAP |
|---|---|---|---|---|---|---|---|---|---|---|
| MNIST | $M_s$ | 0.39 | 0.20 | 0.39 | 0.17 | 0.28 | **0.51** | 0.26 | 0.15 | 0.43 |
| | $1-M_\sigma$ | 0.95 | **1.00** | 1.00 | 0.96 | 0.99 | 0.99 | 0.93 | 0.99 | 0.99 |
| | $M_{ct}$ | 0.65 | 0.70 | 0.69 | 0.67 | **0.71** | 0.70 | 0.69 | 0.69 | 0.70 |
| Fashion MNIST | $M_s$ | 0.61 | 0.40 | 0.61 | **0.66** | 0.60 | 0.65 | **0.66** | 0.11 | **0.66** |
| | $1-M_\sigma$ | 0.96 | **1.00** | 0.99 | 0.96 | 0.99 | 0.99 | 0.93 | 0.99 | 0.99 |
| | $M_{ct}$ | 0.70 | 0.75 | 0.82 | **0.85** | 0.80 | 0.82 | 0.81 | 0.45 | 0.82 |
| COIL-20 | $M_s$ | 0.52 | **0.65** | 0.18 | 0.30 | 0.39 | 0.25 | 0.21 | 0.03 | 0.39 |
| | $1-M_\sigma$ | 0.99 | 0.99 | 1.00 | 0.99 | 1.00 | 1.00 | 0.97 | 0.99 | **1.00** |
| | $M_{ct}$ | **0.74** | 0.73 | 0.60 | 0.65 | 0.69 | 0.58 | 0.63 | 0.55 | 0.70 |
| Cifar10 | $M_s$ | 0.65 | 0.24 | 0.68 | 0.74 | 0.72 | **0.78** | 0.61 | 0.88 | 0.74 |
| | $1-M_\sigma$ | 0.99 | **1.00** | **1.00** | 0.98 | **1.00** | **1.00** | 0.97 | 0.97 | **1.00** |
| | $M_{ct}$ | **0.93** | 0.67 | 0.92 | 0.91 | 0.92 | 0.92 | 0.86 | 0.90 | **0.93** |
| RNA-seq | $M_s$ | 0.44 | 0.18 | 0.39 | 0.17 | 0.28 | 0.50 | **0.67** | 0.28 | 0.65 |
| | $1-M_\sigma$ | 0.95 | **1.00** | 1.00 | 0.96 | 0.99 | 0.99 | 0.73 | 0.99 | 0.99 |
| | $M_{ct}$ | 0.74 | 0.60 | 0.79 | **0.84** | 0.79 | 0.79 | 0.76 | 0.90 | 0.80 |

(b) Global metrics

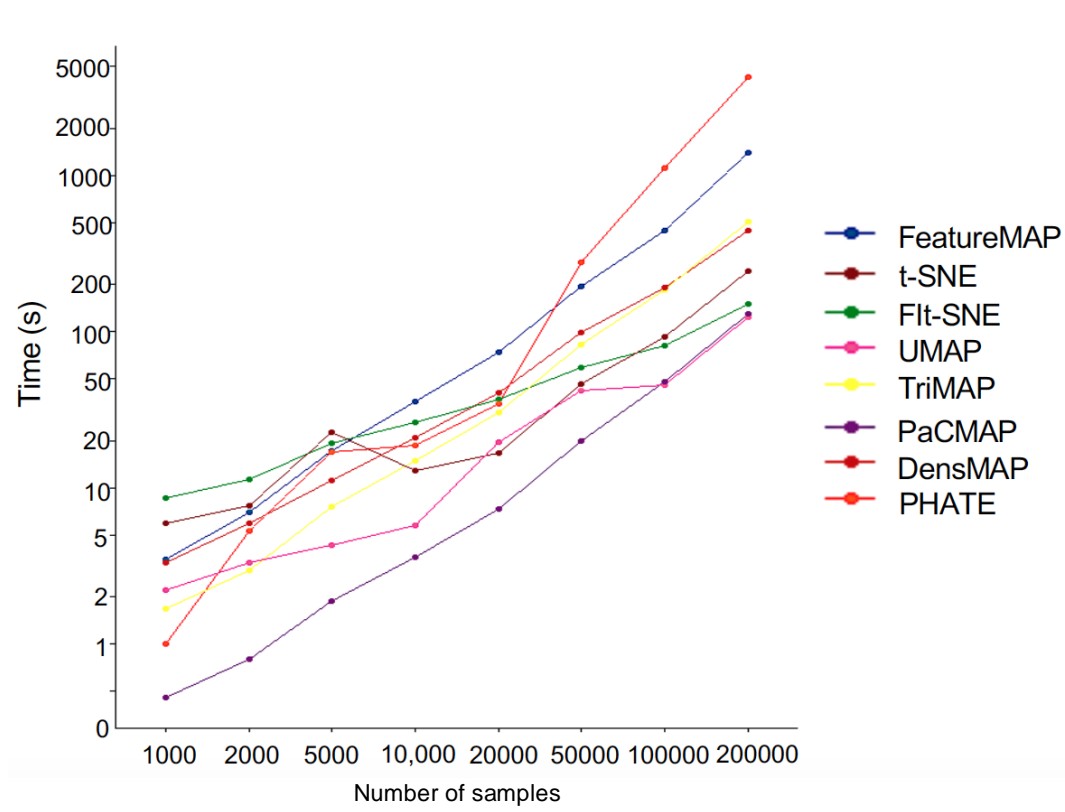

Figure 14: Comparison of running time across different sample sizes. FeatureMAP is comparable with other methods. The gap between FeatureMAP and UMAP is from both local PCA calculation and anisotropic projection.

### A.2.5 FeatureMAP enables visualizing feature directions and importance by feature gradients

In this section, we demonstrate FeatureMAP's ability to preserve and visualize important features of the source space, which methods like UMAP and t-SNE fail to achieve in Fig. 15.

Unlike non-linear methods, the linear PCA can illustrate feature contributions through a biplot, where feature contributions explain how source features influence the data distribution. In Fig. 15a, we visualize feature contributions in a PCA plot. Specifically, the selected feature, *pixel127*, is highlighted with its feature contribution and feature count. The feature contribution and feature count are consistently correlated, as the arrow of the feature contribution points toward the region with a higher feature count.

In contrast, the nonlinear methods UMAP and t-SNE are devised such that no source features are retained in the embedding. Their embedding, especially the two-dimensional visualization, are coordinates to maintains the relative positions in high-dimensional space. Since these coordinates are the only available information in the embedding, we mimic PCA's biplot to associate source features with UMAP and t-SNE's embedding by computing the correlation of features with the embedding coordinates. For each feature $f$ in high-dimensional space, its correlations with the embedding coordinates $y_1$ and $y_2$ are $corr_1 = \frac{<f,y_1>}{||f||||y_1||}$ and $corr_2 = \frac{<f,y_2>}{||f||||y_2||}$, respectively. These two correlations form an arrow for the feature in the embedding, illustrating how the feature interprets the embedding (Fig. 15b, c).

Our FeatureMAP enhances nonlinear dimensionality reduction by introducing feature-preserving functionality. Based on the manifold's assumption that data points are locally distributed over Euclidean space, we compute PCA locally to derive feature gradients, representing local feature contributions in terms of both direction and magnitude (Fig. 15d). To ensure consistency between local feature contributions and the data embedding, we first align the local gauges by projecting the tangent space into the low-dimensional space (section 3.3). This alignment allows local feature contributions to be effectively revealed through the local gauges.

To further benchmark FeatureMAP's feature-preserving functionality with PCA, UMAP and t-SNE, we show the 10 most important features identified by each method in terms of feature contribution, feature importance and feature count Figs. 16 to 19. FeatureMAP outperforms the others on visualizing features. Therefore, FeatureMAP combines the strengths of manifold learning and principal component analysis, augmenting UMAP with feature-preserving properties.

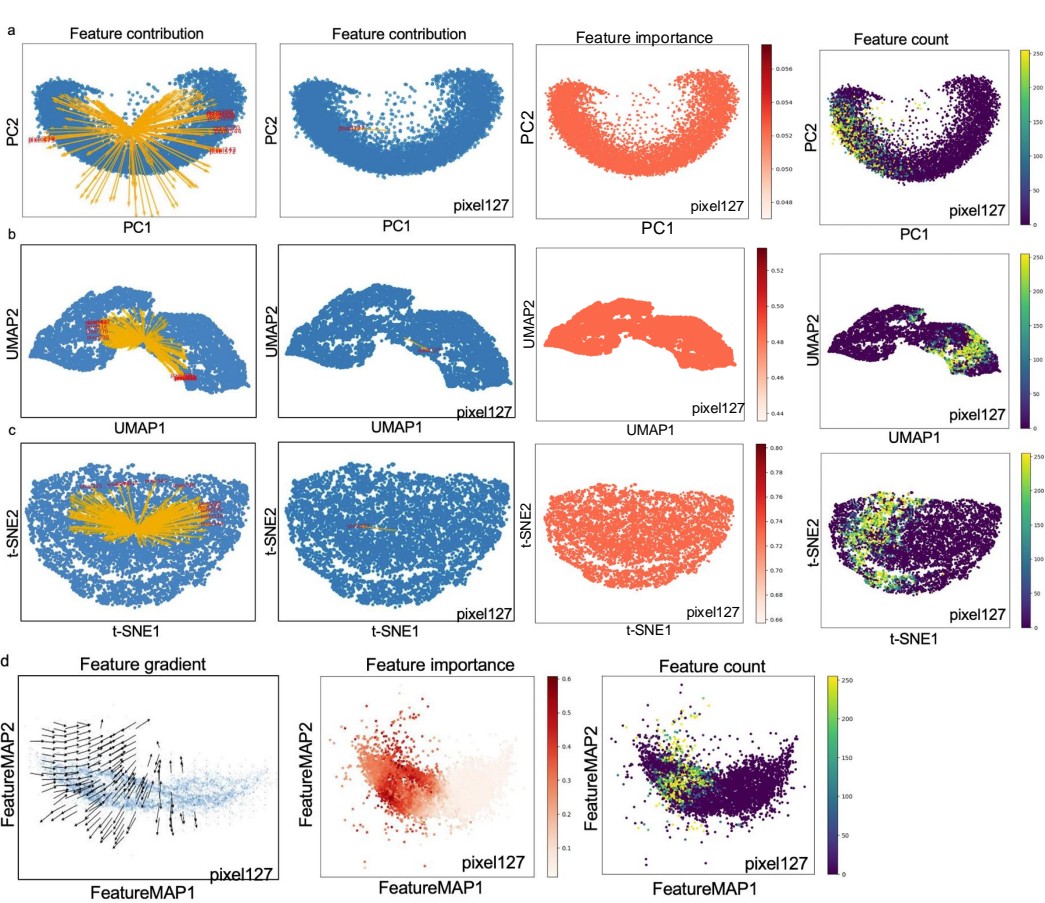

Figure 15: FeatureMAP enables the visualization of feature directions and importance using feature gradients. a, b, c. Biplots display feature contributions along data embeddings for PCA, UMAP, and t-SNE. Top important features are labeled (left). A selected top feature, *pixel127*, is shown with its feature contribution, feature importance and feature count. d. FeatureMAP embedding visualizes feature gradients, highlighting feature directions and importance.

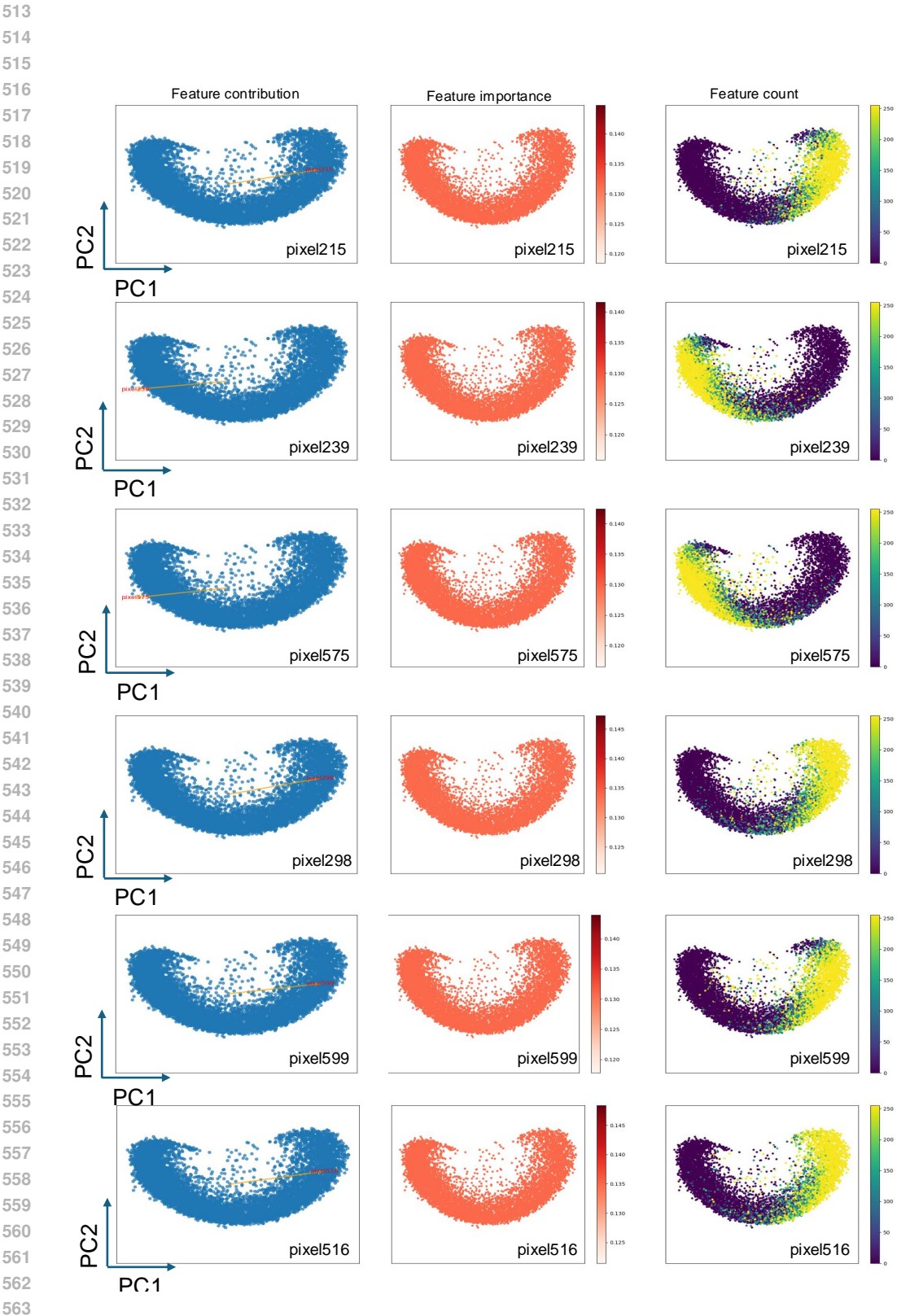

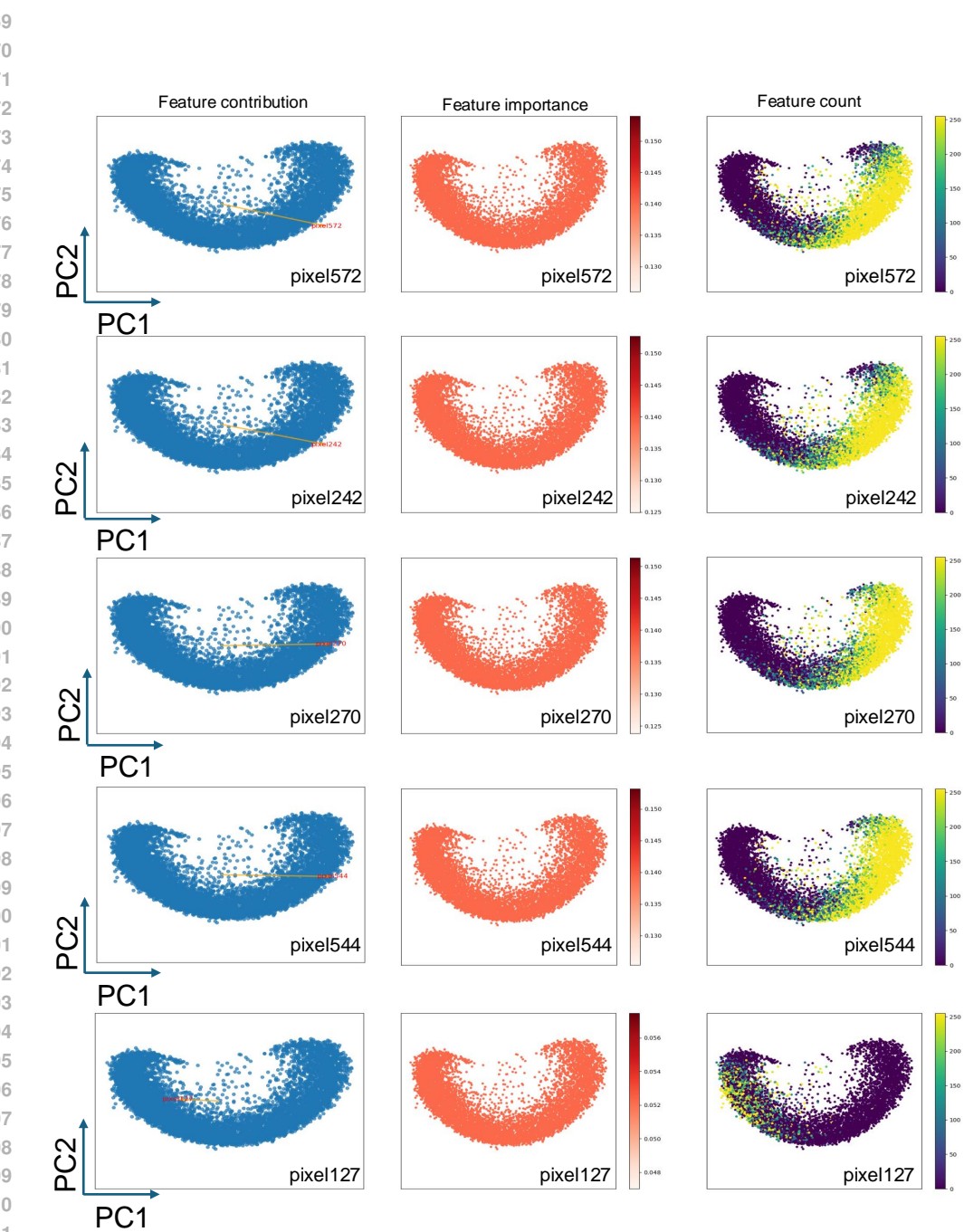

Figure 16: PCA shows the feature contribution, feature importance and feature count.

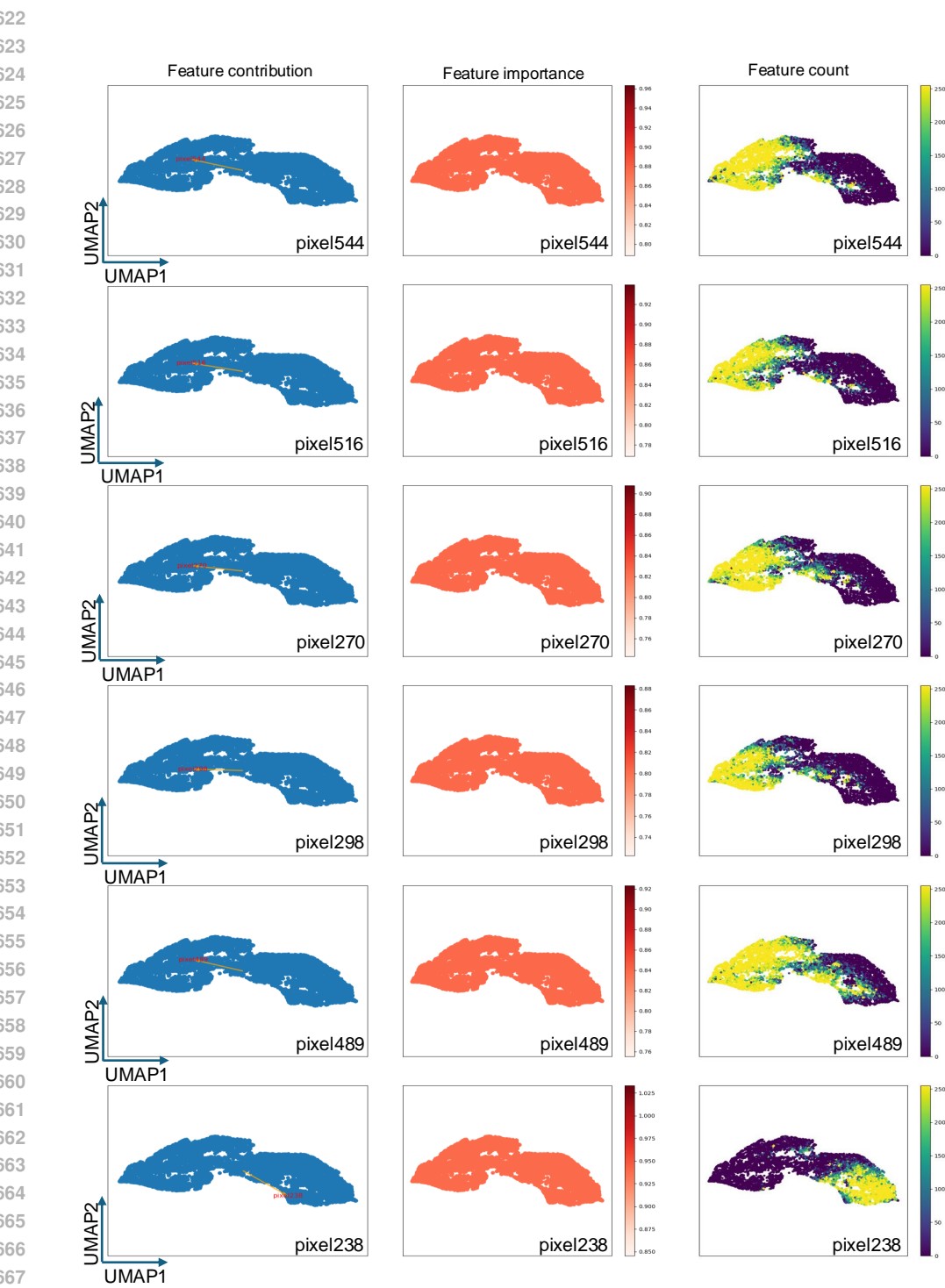

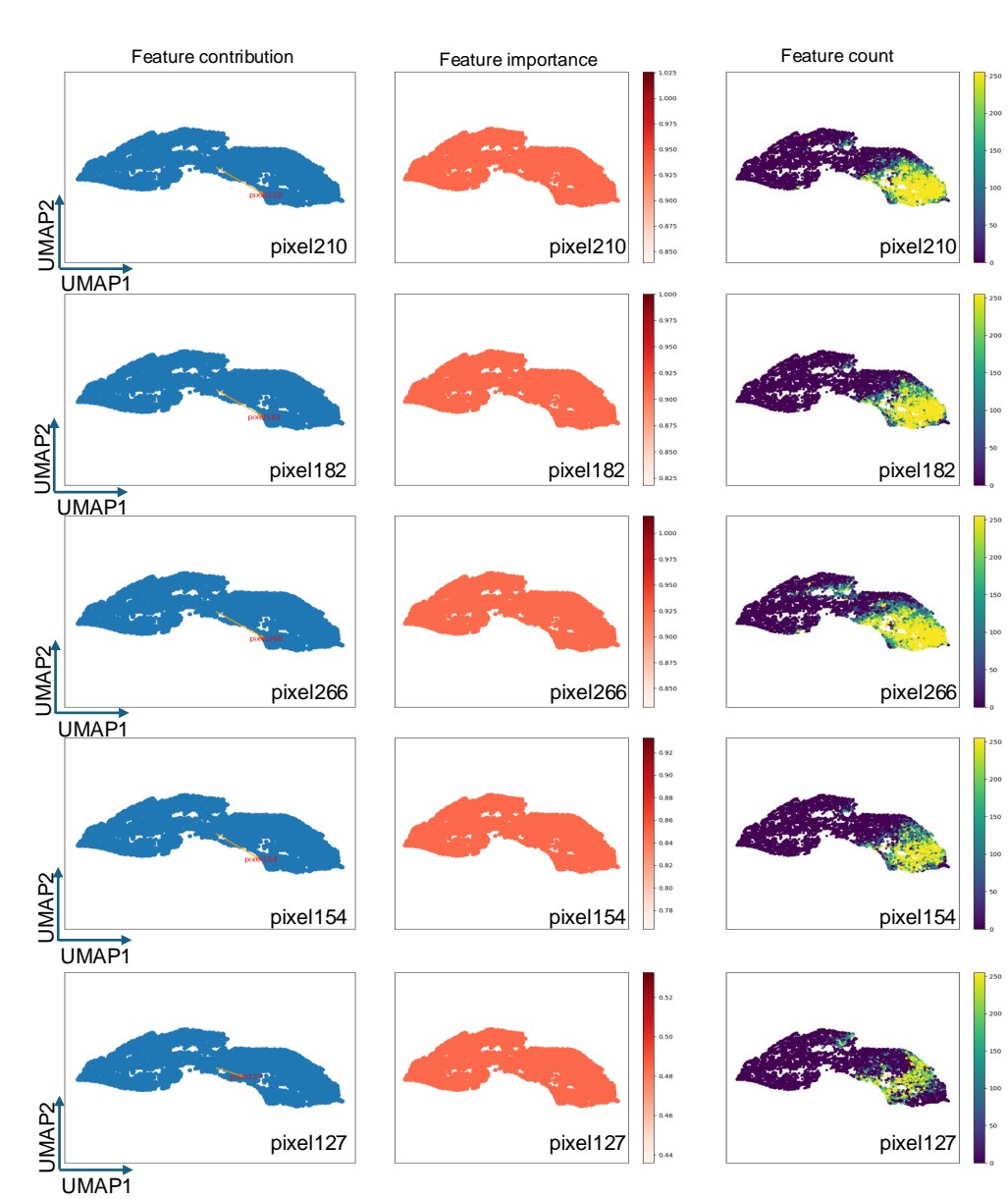

Figure 17: UMAP shows the feature contribution, feature importance and feature count.

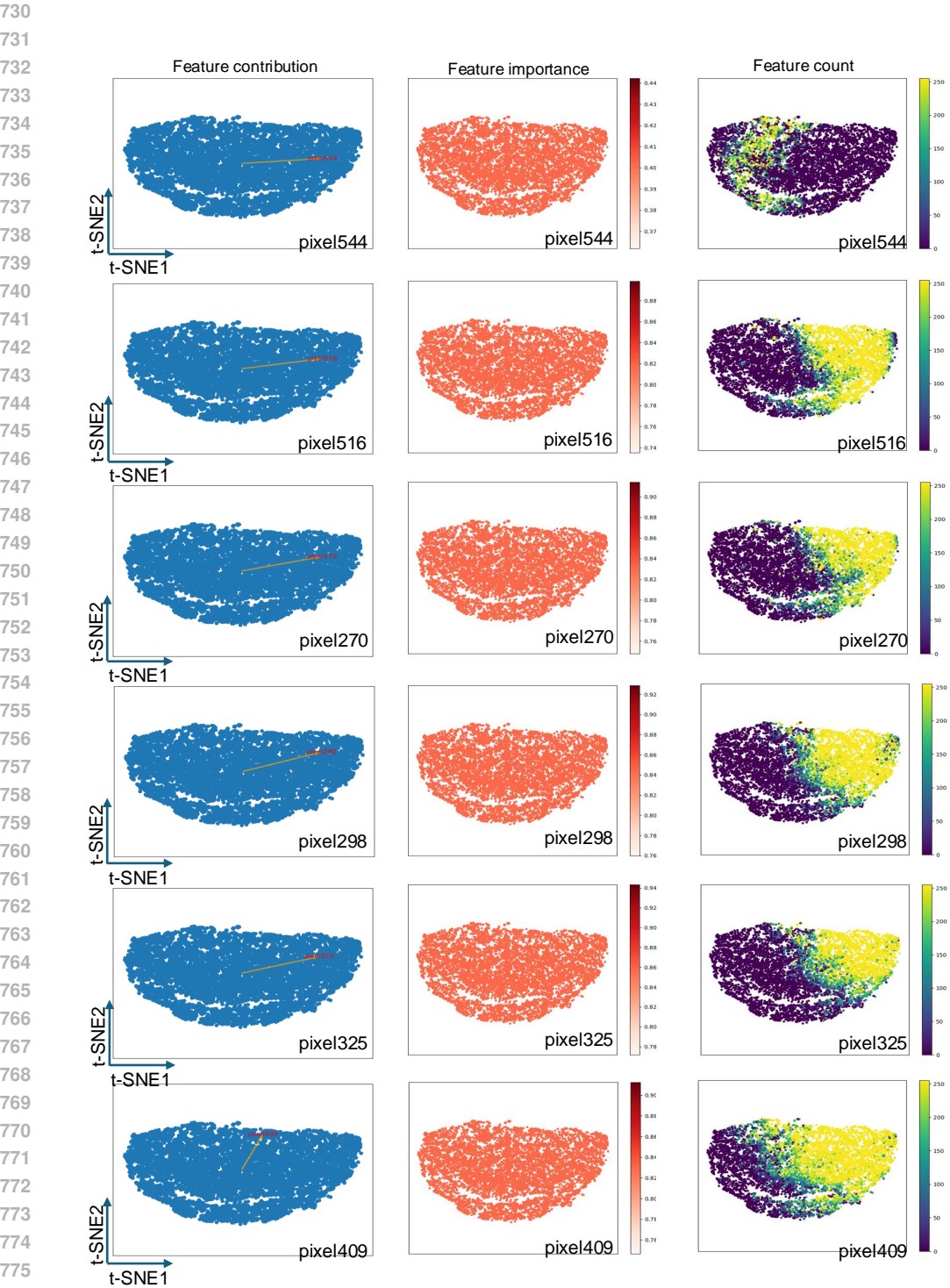

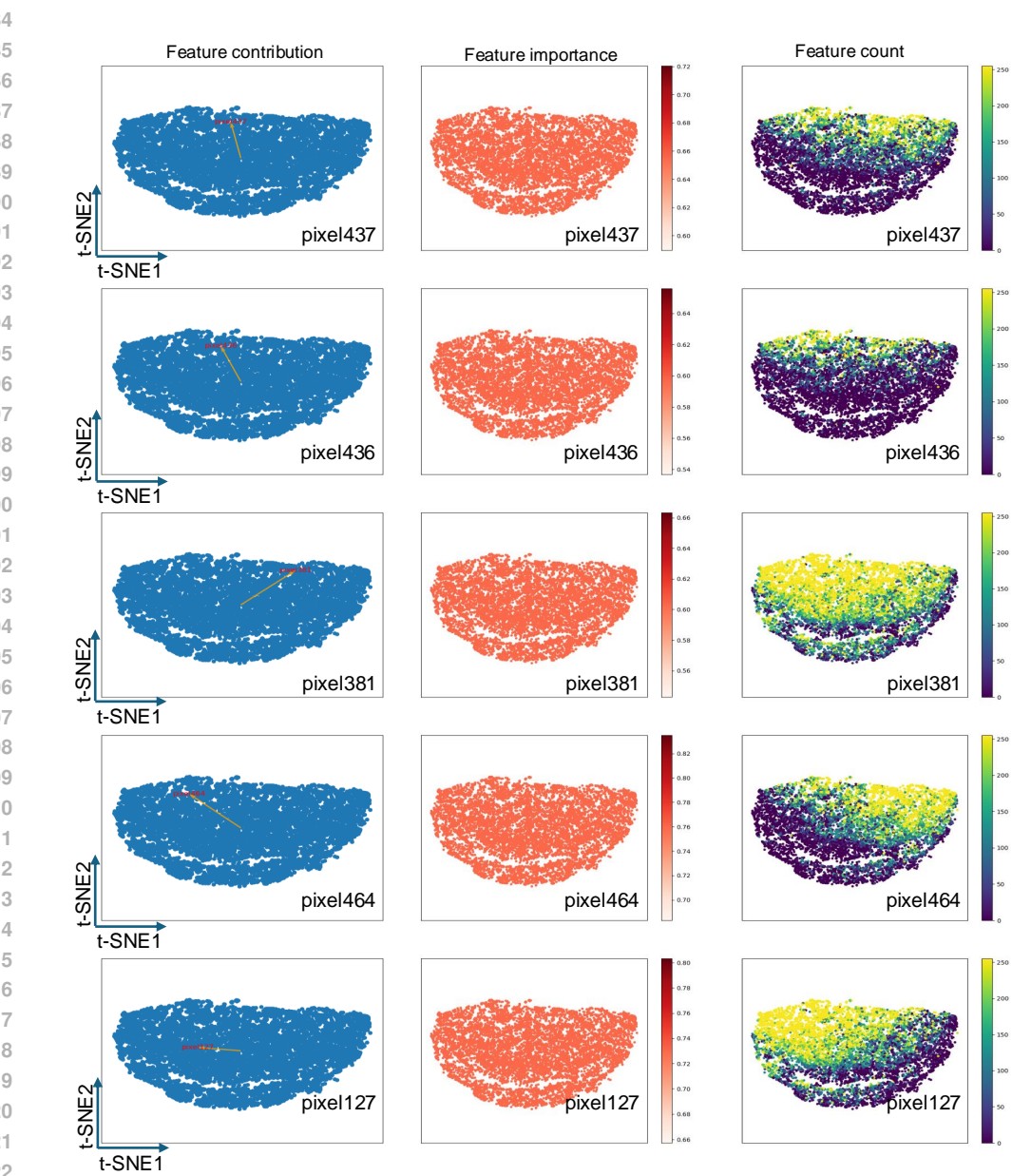

Figure 18: t-SNE shows the feature contribution, feature importance and feature count.

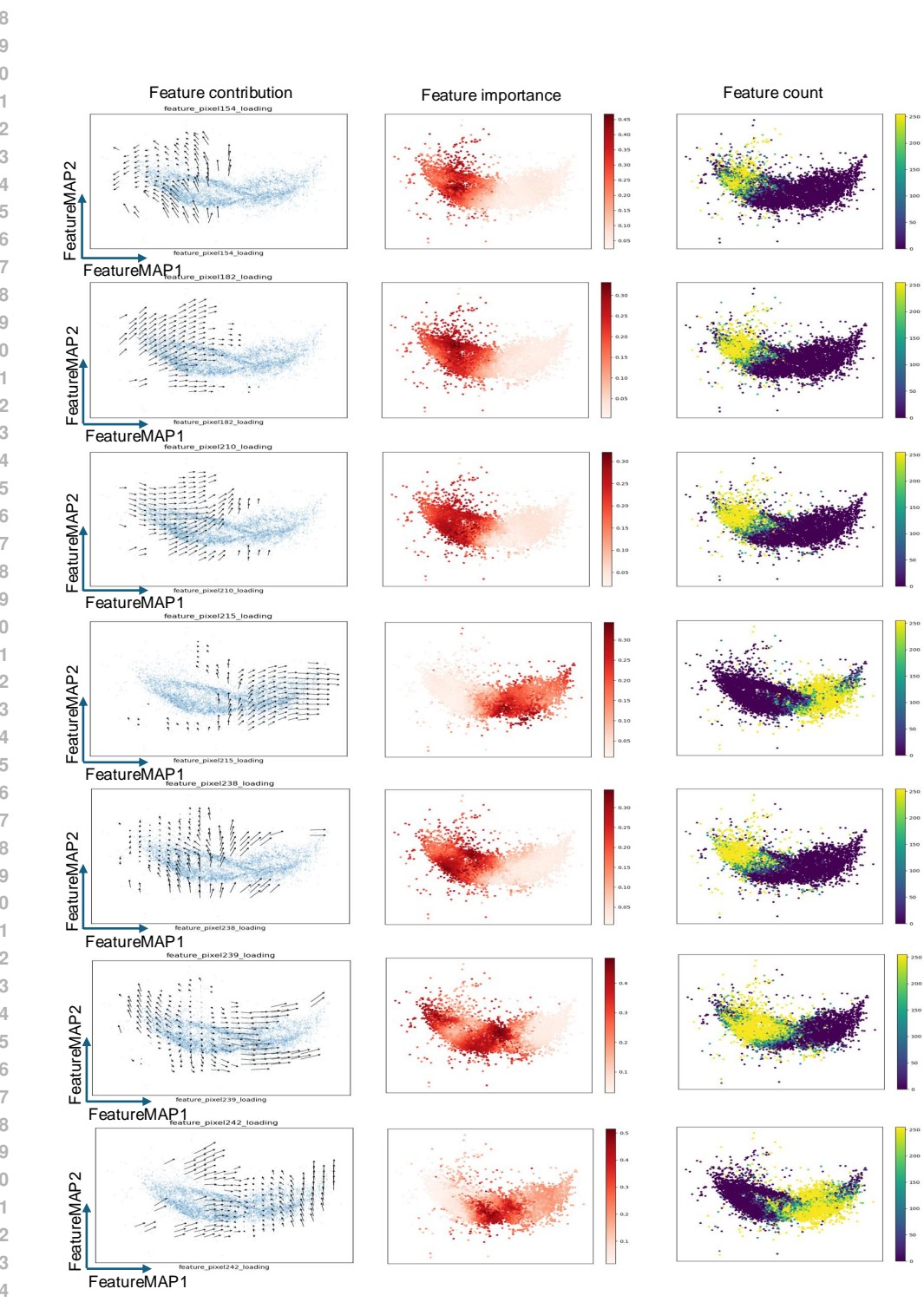

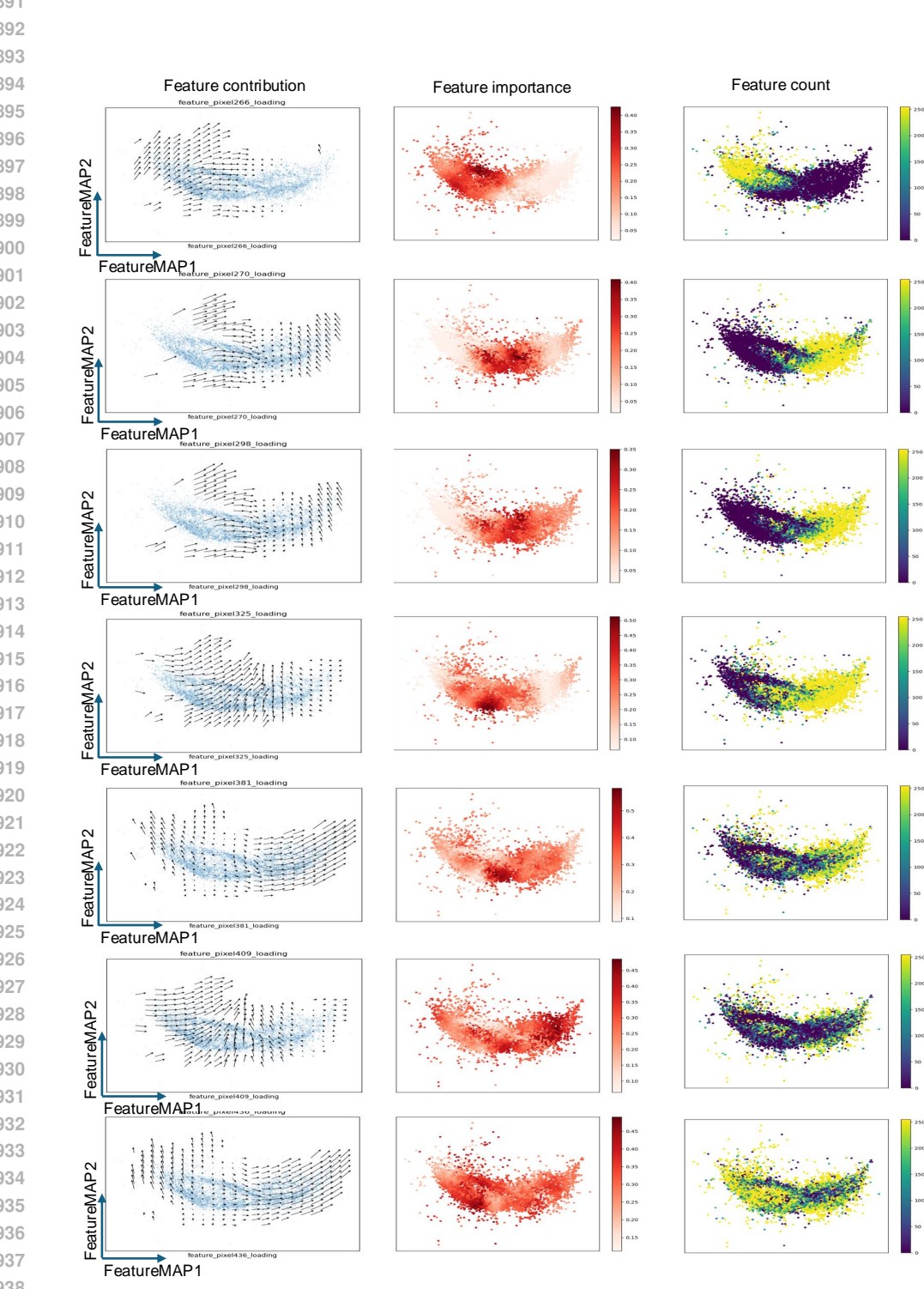

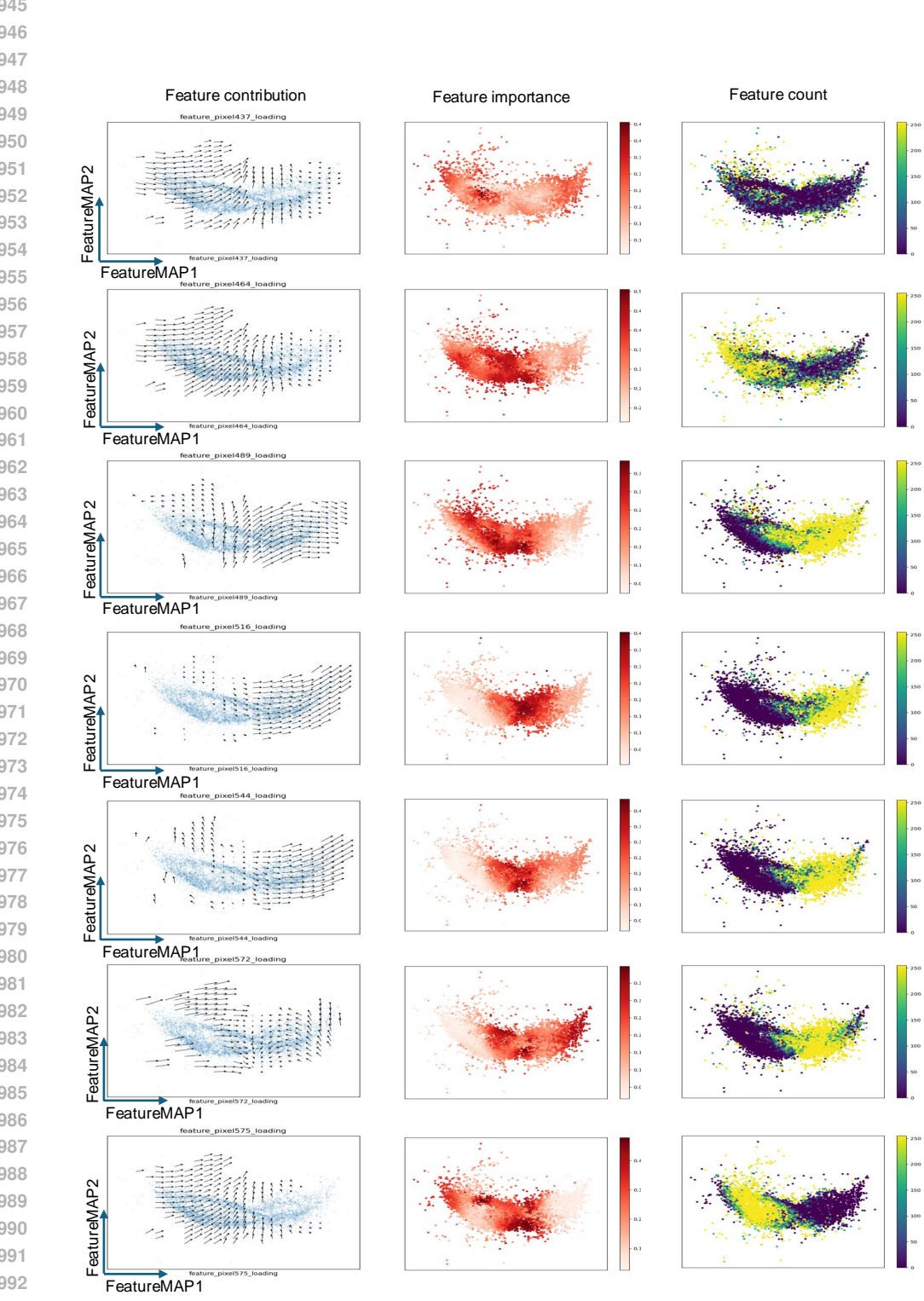

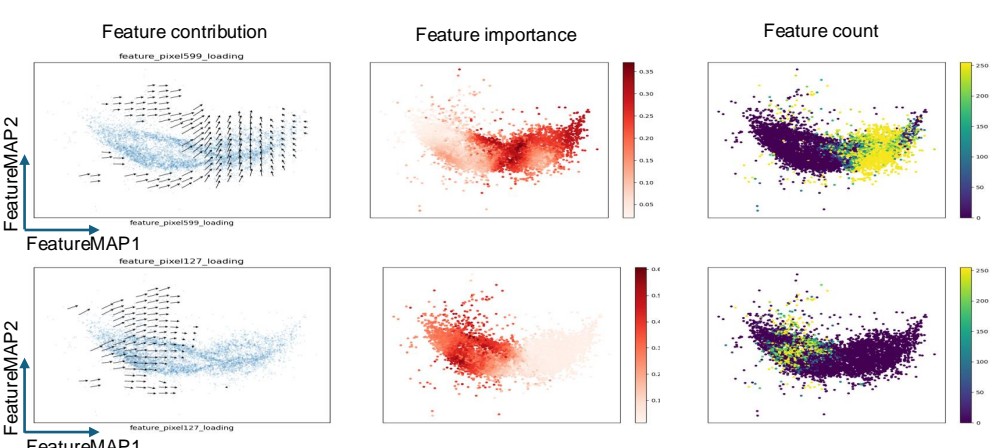

Figure 19: FeatureMAP shows the feature contribution, feature importance and feature count.

