# OpenReview forum: "Interpretable Dimensionality Reduction by Feature-preserving Manifold Approximation and Projection"
_ICLR.cc/2025/Conference — Submitted to ICLR 2025_

### Official Review · Reviewer_abkL · 2024-10-30

**Soundness:** 2
**Presentation:** 3
**Contribution:** 2
**Rating:** 5
**Confidence:** 4

**Summary:**

Nonlinear dimensionality reduction is a classic research area in past years. In the paper, the authors proposed a feature-preserving manifold approximation and projection method, called FeatureMAP. The proposed method employs anisotropic projection to preserve both the manifold structure and the original data density. Some methods are compared with the proposed method on FashionMNIST and COIL-20 dataset to demonstrate the effectiveness of the method.

**Strengths:**

1. In the paper, the authors proposed a feature-preserving manifold approximation and projection method, called FeatureMAP.
2. Some methods are compared with the proposed method on FashionMNIST and COIL-20 dataset to demonstrate the effectiveness of the method.

**Weaknesses:**

1. Unclear Motivation: Why perform SVD on $ W_i X_i $ instead of Xi or X? The motivation is not introduced clearly.
2. There is an error in (5), since Vi/Vj is known variable calculated in (2).
3. What are the advantages of the method compared with the existing works, especially in terms of computational complexity, running times, and robustness?
4. What’s the significance/application value of the method in real-word applications, especially under the LLM era?
5. This method involves multiple independent steps and is not a complete optimization problem. Will this lead to difficulties in obtaining the global optimal solution and the problem of the previous step affecting the next result?

**Questions:**

see weakness

---

> ### Author Response · Authors · 2024-11-21
>
> >1 Unclear Motivation: Why perform SVD on instead of Xi or X? The motivation is not introduced clearly.
>
> Thank you for your question. The motivation for performing SVD on $W_iX_i$ rather than $X_i$ lies in capturing the local structure of the data. $W_i$ applies weights to emphasize the local neighborhood of each data point $x_i$ , ensuring that the SVD captures the most significant local variation. This approach provides a more accurate tangent space approximation by considering the local geometry, which would be lost if unweighted $X_i$ were used. We have clarified this motivation in the revised manuscrip, which is highlighted in yellow in section 3.1.
>
> >2 There is an error in (5), since Vi/Vj is known variable calculated in (2).
>
> Thank you for you suggection. The objective of Eq (5) is to compute the optimal alignment $O_{ij}$, which is an orthonormal matrix ($\textit{O}(d)$), given the tangent space $V_i$ and $V_j$. Thus, $O_{ij}$ is regarded as an approximation of $V_jV_i^T$. We have clarified this point further in the manuscript highlighted in yellow in section 3.2.
>
> >3 What are the advantages of the method compared with the existing works, especially in terms of computational complexity, running times, and robustness?
>
> Thank you for your question. FeatureMAP provides several distinct advantages over existing methods, particularly with its unique focus on feature preservation, alongside considerations of computational complexity, running times, and robustness:
>
> Feature Preservation: Unlike many dimensionality reduction methods, such as UMAP or PHATE, FeatureMAP explicitly embeds tangent spaces, allowing it to preserve feature-level relationships in addition to the manifold structure. This makes FeatureMAP highly interpretable, as it highlights which features contribute most to the embedding, a capability lacking in most existing methods.
>
> Computational Complexity and Running Times: While FeatureMAP involves additional steps, such as tangent space computation and alignment, it achieves comparable running times to methods like densMAP and TriMAP by leveraging approximate nearest neighbor searches and efficient optimization techniques (e.g., stochastic gradient descent). For datasets of 50,000 samples, FeatureMAP requires slightly more time than UMAP (150 seconds vs. 30 seconds) due to its enhanced functionality, but this is justified by its ability to preserve both feature importance and manifold geometry.
>
> Robustness: FeatureMAP is robust to variations in data density and structure, as its tangent space alignment and embedding steps balance local and global structural preservation. By aligning tangent spaces through cosine similarity and optimizing embeddings to maintain agreement with the original geometry, FeatureMAP ensures stable and consistent results, even for complex datasets.
>
> We have included a comparative analysis of the major existing methods in Section A.1.7 of the APPENDIX.
>
> >4 What’s the significance/application value of the method in real-word applications, especially under the LLM era?
>
> Thank you for this insightful question. FeatureMAP provides substantial value in real-world applications, particularly in the era of large language models (LLMs), where interpretability and efficiency are essential:
>
> - Interpretability of High-Dimensional Representations: FeatureMAP preserves and visualizes feature-level importance in high-dimensional data. It helps uncover the key features or dimensions contributing to downstream tasks, enhancing interpretability in applications such as sentiment analysis, question answering, and content generation.
>
> - Feature Understanding in Fine-Tuning and Evaluation: FeatureMAP identifies which features are preserved, transformed, and lost during LLM fine-tuning. This insight aids domain adaptation and improves model generalization by enabling better understanding of feature interactions.
>
>
> >5 This method involves multiple independent steps and is not a complete optimization problem. Will this lead to difficulties in obtaining the global optimal solution and the problem of the previous step affecting the next result?
>
> Thank you for this important question. FeatureMAP addresses the challenges of combining multiple independent steps by using a unified loss function that jointly optimizes kNN graph embedding and anisotropic projection based on tangent space embedding. This approach minimizes the risk of suboptimal results affecting subsequent steps.
>
> The loss balances two key objectives: kNN Graph Embedding and  Tangent Space Alignment and Projection.
> By optimizing these objectives simultaneously, the method aligns local and global structures. The use of stochastic gradient descent (SGD) iteratively minimizes the combined loss, ensuring convergence to a solution that integrates graph embedding and tangent space alignment effectively.
>
> We have revised the algorithm pseudocode in Section A.1.6 to enhance clarity.

---

> > ### Comment · Reviewer_abkL · 2024-11-26
> >
> > Thanks to the authors for the responses and revision.
> > Since the authors emphasize robustness and local information, can Wi ensure completely correct neighbor information? Can it ensure that non-similar neighbor information is not introduced, especially when there is noise? The authors mentioned that Feature Understanding in fine-tuning for LLM, the output of the large model should be the feature vector directly. How to understand the feature understanding in the application of LLM? Can you give some examples?

---

> > > ### Author Response · Authors · 2024-11-26
> > >
> > > Thank you for your valuable and insightful comments.
> > >
> > > > Since the authors emphasize robustness and local information, can $W_i$ ensure completely correct neighbor information? Can it ensure that non-similar neighbor information is not introduced, especially when there is noise?
> > >
> > > Thank you for raising this important question.
> > > The weights $W_i = diag(\sqrt{P_{i1}} , ..., \sqrt{P_{ik}} )$ in our method are derived from the approximate kNN graph computation [1] and UMAP's local probability definition:
> > >
> > > $
> > > P_{j|i} = \exp\left(-\frac{\|x_i - x_j\| - \text{dist}_i}{\gamma_i}\right),
> > > $
> > >
> > > where  $x_j$ is within the kNN neighbours of $x_i$,  $dist_i$ is the local distance normalization term, and $\gamma_i $controls the spread of the local neighborhood. This computation ensures a probabilistic interpretation of neighbor relationships based on distance.
> > >
> > > To construct the kNN graph, we use an approximate kNN search algorithm with guaranteed accuracy bounds, as detailed in the work by Dong et al. (2011) [1]. This algorithm balances computational efficiency and accuracy, minimizing the risk of introducing non-similar neighbors.
> > >
> > > While no kNN method is completely immune to noise, the approximate kNN algorithm we use provides theoretical guarantees and has been experimentally demonstrated to retrieve true positives effectively while controlling false positives, as shown in Dong et al. (2011). This ensures the reliability of $W_i $ in capturing meaningful local structures, even in the presence of noise.
> > >
> > > We have revised our manuscript in Section 3.1 (highlighted in yellow) to discuss the accuracy of kNN graph construction, particularly in handling non-similar neighbor information and noise.
> > >
> > > *[1] Dong, Wei, Charikar Moses, and Kai Li. "Efficient k-nearest neighbor graph construction for generic similarity measures." Proceedings of the 20th international conference on World wide web. 2011.*
> > >
> > >
> > >
> > > >The authors mentioned that Feature Understanding in fine-tuning for LLM, the output of the large model should be the feature vector directly. How to understand the feature understanding in the application of LLM? Can you give some examples?
> > >
> > > Thank you for raising this thoughtful question.
> > > Feature understanding in the context of large language models (LLMs) involves interpreting how specific features in LLMs's output feature vectors contribute to specific tasks. FeatureMAP can facilitate this understanding through data visualization and feature visualization using feature gradients. Below, we provide an example of Semantic Similarity and Clustering in LLM applications to illustrate this.
> > >
> > > **Example: Semantic Similarity and Clustering in LLMs**
> > >
> > > When LLMs generate feature vectors (e.g., sentence embeddings), these vectors encode semantic information. FeatureMAP can be used to visualize these embeddings in a lower-dimensional space while preserving important feature-level information.
> > >
> > > 1. **Data Visualization**
> > >
> > > Using FeatureMAP, sentence embeddings are projected into a 2D space, revealing clusters that represent semantically similar sentences. For example, sentences about "technology" may form one cluster, while sentences about "healthcare" form another.
> > > The clustering reveals the relative positioning of embeddings while preserving their local and global structures.
> > >
> > > 2. **Feature Visualization**:
> > >
> > > Feature gradients computed by FeatureMAP highlight the contribution of specific dimensions (features) in the embedding to the clusters.
> > > For instance, if a particular feature represents the presence of technology-related terms, the feature direction points toward the "technology" cluster, and its magnitude indicates the feature's importance.
> > >
> > > Through this process, FeatureMAP helps uncover which features influence clustering and similarity in the embedding space, providing actionable insights for fine-tuning. For example, if a feature critical to a cluster is weakly represented in the embeddings, it may indicate a need for additional fine-tuning of the LLM.
> > >
> > > In addition, expanding FeatureMAP's capabilities to enhance feature interpretability in LLMs will be a focus of our future work, as we believe it can improve the interpretability in LLM applications.

---

> > > > ### Author Response · Authors · 2024-11-30
> > > >
> > > > Dear Reviewer,
> > > >
> > > > Thank you for your time and efforts in reviewing our submission. We have carefully addressed your comments and submitted our revised manuscript through the system. We kindly request you to review the rebuttal at your earliest convenience.
> > > >
> > > > If there are any additional clarifications or details we can provide, please do not hesitate to let us know. We appreciate your valuable feedback and support in this process.

---

### Official Review · Reviewer_3oL1 · 2024-11-02

**Soundness:** 3
**Presentation:** 2
**Contribution:** 3
**Rating:** 6
**Confidence:** 4

**Summary:**

Nonlinear dimensionality reduction often lacks interpretability due to the absence of source features in low-dimensional embedding space. This paper proposes FeatureMAP, an interpretable method that preserves source features by tangent space
embedding.

**Strengths:**

This paper proposes FeatureMAP, an interpretable method that preserves source features by tangent space embedding. The core of FeatureMAP is to use local principal component analysis (PCA) to approximate tangent spaces.

**Weaknesses:**

The way to solve Eq. (11) is not described in detail. A.4 seems hard for me to implement the FeatureMAP.

**Questions:**

see the weaknesses

---

> ### Author Response · Authors · 2024-11-21
>
> >Weaknesses:
> The way to solve Eq. (11) is not described in detail.
>
> Thank you for raising this question. We have expanded the description of the computation for Eq. (11) in Section A.1.4 of the APPENDIX by including details of the SGD calculation. The updated section is highlighted in blue.
>
>
> > A.4 seems hard for me to implement the FeatureMAP.
>
> Thank you for your question. We have revised the pseudocode for FeatureMAP in Section A.1.6 of the APPENDIX to enhance readability. The main algorithm is now divided into seven steps, with each step presented in its own pseudocode.

---

> > ### Comment · Reviewer_3oL1 · 2024-11-25
> >
> > Thanks for your responses. I decide to keep the score.

---

### Official Review · Reviewer_57HL · 2024-11-03

**Soundness:** 2
**Presentation:** 3
**Contribution:** 2
**Rating:** 5
**Confidence:** 3

**Summary:**

The manuscript suggests a new non-linear dimensionality reduction method, which aims to preserve important features of the original space by preserving the alignment between the local axes ("tangent space") defined at each sample. It can be seen as an extension to UMAP, where the data topology is defined through kNN, and the loss function is optimised through gradient descent. The loss function has two terms: one which minimises distortion of distances caused by the projections (like UMAP) and a second term which maximises correlations of the radii of the local axis (the singular values of the data directions around each data point). This optimisation of the alignment of the local axes aims to preserve both the global features of the source space and the data density on the low-dimensional manifold.

**Strengths:**

- the manuscript provides interesting intuition for why the alignment between local axes of variation needs to be preserved by nonlinear dimensionality reduction methods.
 - the manuscript shows how to extend UMAP to include the alignment of local axes in the loss function in a way which can still be optimised in practice.
 - experimental results show how the new method differs from previously proposed ones in terms of the resulting visualisation and data density preservation.

**Weaknesses:**

- little or no support is provided that the resulting dimensionality reduction better preserves important features of the source space, which is the main motivation of the method.
 - the comparison to other methods is mostly qualitative and makes it hard to argue the new method is somehow superior to previously proposed ones. Visualization results (Figure 12) are subjective ("clearly shows distint clusters of different digit groups"), quantitative measures (Table 1) are inconclusive, and running time is usually worse (Figure 13).

**Questions:**

1. Is it a correct assessment to say that the proposed method is equivalent to UMAP when considering only the topology (kNN graph) and the first term of the loss function (distance distortion)? If so, it would clarify the presentation if you would say so and not repeat the parts already part of UMAP (you can, of course, have the full details of the loss in the appendix).
2. The second optimisation term has two interpretations: preservation of local density and preservation of source features. Can you provide evidence that the method better preserves source features than UMAP or other methods?
3. A recent criticism of dimensionality reduction methods and their interpretability in the context of biology  [Chari & Pachter PlosCompBio (2023)] might be relevant to refer too. Would you say the criticism equally applies to your method or would you suggest the proposed method somehow improve the interpretability of such results?

---

> ### Author Response · Authors · 2024-11-22
>
> >Weaknesses:
> little or no support is provided that the resulting dimensionality reduction better preserves important features of the source space, which is the main motivation of the method.
>
> Thank you for this valuable comment. The core motivation for proposing FeatureMAP is its ability to preserve and visualize important features of the source space, which existing methods, such as UMAP and t-SNE, fail to achieve because they only consider embedding the topological structure by kNN graph.  Unlike these methods, FeatureMAP embeds tangent spaces, enabling it to explicitly preserve feature directions and feature importance in the reduced space. This capability allows users to not only visualize the embeddings but also understand which features contribute most to the low-dimensional representation.
> To validate this feature-preservation property, we conducted experiments on the MNIST, Fashion-MNIST, COIL-20 dataset and adversarial examples. FeatureMAP successfully highlights the directions and importance of features, demonstrating its robustness and interpretability in scenarios where understanding the role of individual features is critical.
>
>
> >the comparison to other methods is mostly qualitative and makes it hard to argue the new method is somehow superior to previously proposed ones. Visualization results (Figure 12) are subjective ("clearly shows distint clusters of different digit groups"), quantitative measures (Table 1) are inconclusive, and running time is usually worse (Figure 13).
>
> Thank you for your feedback. FeatureMAP uniquely visualizes feature directions and importance, enabling data interpretation in embedding space, unlike UMAP or t-SNE. This interpretability is our principal contribution.
> - Quantitative Performance:
> FeatureMAP outperforms competitors in local metrics (e.g., continuity, kNN accuracy) and ranks consistently high on global metrics (e.g., Shepard goodness, stress), as shown in Table 1.
> - Computational Cost:
> FeatureMAP’s higher runtime (e.g., 150s vs. UMAP’s 30s) is justified by its extended functionality, including tangent space embedding and feature visualization, which surpass solely kNN-based methods.
> - Visualization and Density:
> FeatureMAP preserves density like densMAP while producing distinct clusters. These visualizations (Figure 12) align with superior quantitative performance, reinforcing the method’s effectiveness.
>
> We have further clarified these points in the manuscript to strengthen our arguments and provide additional context to the comparisons in Section A.2.4
>
> >Questions 1
>
> Thank you for raising this concern. Yes, our method FeatureMAP is equivalent to UMAP when considering only the topology (kNN graph) and the first term of the loss function (distance distortion). We have clarified this in the manuscript by appropriately attributing the corresponding techniques to UMAP and have revised Section 3, with updates highlighted in red.
>
> >Questions 2
>
> Thank you for your question. The second optimization term in FeatureMAP preserves both local density and source features, uniquely enabling the visualization of feature directions and importance—capabilities not available in UMAP or t-SNE.
>
> Evidence from the MNIST, Fashion MNIST, COIL-20 datasets, and adversarial examples demonstrates FeatureMAP's ability to highlight feature directions and their influence in the embedding space. Furthermore, as shown in Figures 8, 9, and 10, FeatureMAP outperforms UMAP in density preservation.
>
> >Questions 3
>
> Thank you for raising this concern. We have added a reference to [Chari & Pachter, PLOS Comput. Biol. (2023)] and discussed its relevance to our method in Related Work (Section 2).
>
> It is worth noting the ongoing debate around UMAP and t-SNE in visualization applications, highlighted both in research publications and on social media. On one side, [Chari & Pachter, PLOS Comput. Biol. (2023)] argue that such embeddings can be arbitrary and misleading, comparing them to forcing data into an elephant shape. On the other side, studies like [Lause et al., PLOS Comput. Biol. (2024)], [Damrich et al., ICLR (2022)], and [Van Assel et al., NeurIPS (2022)] show that while these methods do not preserve high-dimensional distances, they can still provide biologically meaningful insights through spectral clustering and probabilistic graph coupling analyses.
>
> Our method aligns with the latter perspective. Specifically, FeatureMAP extends conventional UMAP by incorporating local tangent space embedding, thus improving the interpretability in terms of both feature importance and density. This augmentation provides new functionality with feature direction and importance to better interpret the embedding. Also, it preserves the denstiy, which enhances the understanding the local and global embedding.

---

> > ### Comment · Reviewer_57HL · 2024-11-22
> > **Response to authors comment**
> >
> > Thank you for providing those clarifications and answering my questions, and I much appreciate your effort to use them to improve the manuscript. I believe you provide good answers to my questions, except for the first (and perhaps main) concert:
> >
> > Where do you show that the new method's `ability to preserve and visualize important features of the source space, which existing methods, such as UMAP and t-SNE, fail to achieve`. What are the source space features you preserve, and where can I see them?
> >
> > Don't get me wrong, I agree with your intuition, and it's a step in the right direction, but it is hard for me to see if your method is doing what your intuition suggests it should be doing. The supporting evidence is not convincing to me, but I am open to be convinced otherwise.

---

> ### Author Response · Authors · 2024-11-23
>
> Thank you so much for your prompt response, and thank you again for raising this important question. To address your concern, we have revised the manuscript to include experiments and figures in Section A.2.5 of the APPENDIX (highlighted in red), illustrating FeatureMAP's feature-preserving strength compared to UMAP, t-SNE, and PCA.
>
> Briefly, the source features we aim to preserve are analogous to the feature loadings in PCA's biplot, which explain a feature's contribution to the data distribution.
> However, this interpretation is not directly applicable to nonlinear dimensionality reduction methods like UMAP and t-SNE, as they do not provide feature loadings in the embedding.
> FeatureMAP addresses this limitation by introducing the computation of tangent spaces and feature gradients over the manifold,
> thus making the feature loadings avaiable to locally interpret the embedding.
>
> Specifically, FeatureMAP is able to visualize the feature contribution by vector filed (Figure 1f in Section 1 and Figure 15 in Section A.2.5), where the feature direction indicates how the feature changes along the data embedding. In addition, the magnitude of feature contribution, i.e., feature importance, directly indentify the key features for each sample, with applications on revealing edges of MNIST, Fashion MNIST and COIL-20, and explaining the misclassification of advarserial attack. Moreover, due to the feature-preserving machanism, the embedding also preserve the data density because we model the local data points in one hyperecllipse.
>
> In summary, source features in this dimensionality reduction context represent feature contributions to the data embedding in terms of both direction and magnitude. FeatureMAP achieves this through feature gradients, providing practical applications in visualizing feature directions and leveraging feature importance to interpret embeddings.
>
> We hope this updated revision addresses your concern. Please feel free to share any additional comments if there are further issues.

---

> > ### Comment · Reviewer_57HL · 2024-11-25
> > **Response to authors**
> >
> > I appreciate the authors' effort in answering my questions; I would have expected Figure 15, which tries to support the central claim that "FeatureMAP enables the visualization of feature directions and importance using feature gradients." to be part of the main text, but nevermind at this stage. I wish you could show more of that:
> >
> >  * You present a single cherry-picked feature...
> >  * The colour map in Figure 16d is not consistent with Figurre16abc, thus making it hard to do a fair comparison.

---

> > > ### Author Response · Authors · 2024-11-26
> > >
> > > Thank you for your comment. We have revised the manuscript based on your suggestions.
> > >
> > > >You present a single cherry-picked feature...
> > >
> > > Apologies for the misleading selection. The reason we chose this feature (i.e., pixel127) was to maintain consistency with Figure 1, where pixel127 is highlighted as the top feature in that example. In our revised manuscript, we focus exclusively on the top 10 important features, visualising them using PCA, UMAP, and FeatureMAP. As the top 10 features identified by each method differ, we combined all these important features, resulting in a set of 22 features illustrated through FeatureMAP.
> > >
> > > >The colour map in Figure 16d is not consistent with Figurre16abc, thus making it hard to do a fair comparison.
> > >
> > > We have added the feature importance panel to ensure consistency in the colormap. In comparison, FeatureMAP highlights heterogeneous feature importance, whereas the other methods only provide an overall score.
> > >
> > > Please feel free to share any additional comments.

---

> > > > ### Comment · Reviewer_57HL · 2024-11-26
> > > > **Response to authors comment**
> > > >
> > > > Thank you for answering that, I decided to increase my score.

---

### Official Review · Reviewer_pkDU · 2024-11-03

**Soundness:** 3
**Presentation:** 3
**Contribution:** 3
**Rating:** 6
**Confidence:** 3

**Summary:**

This paper introduces a novel interpretable nonlinear dimensionality reduction method called FeatureMAP, which aims to enhance the interpretability of reduced-dimensional results by preserving source features. FeatureMAP utilizes local Principal Component Analysis (PCA) to approximate tangent spaces and computes gradients to reveal feature directions and importance. Additionally, it embeds tangent spaces into low-dimensional space while maintaining alignment between them, preserving local anisotropic density. The authors evaluate FeatureMAP on various datasets, and compare it with existing dimensionality reduction methods.

**Strengths:**

1.FeatureMAP offers a new perspective by combining local PCA and tangent space embedding to enhance the interpretability of nonlinear dimensionality reduction, which is a commendable innovation.
2.The paper conducts extensive experiments on several standard datasets, validating FeatureMAP's effectiveness in preserving features and density.
3.The paper effectively demonstrates FeatureMAP's ability to distinguish different categories and explain adversarial samples through visualization results and compares the proposed meathod with some of the most advanced methods, showing its advantages in certain aspects.

**Weaknesses:**

1.The paper mentions the selection of parameter λ but does not discuss the impact of other parameters on the results. Guidance on parameter selection and sensitivity analysis is crucial for practical applications.
2.The paper should discuss the computational complexity of FeatureMAP, especially its performance and scalability when dealing with large datasets. This is a key factor for the feasibility of the method in practical applications.

**Questions:**

1.What is the computational resource consumption of FeatureMAP when dealing with large datasets? Are there optimization strategies to improve efficiency?
2. In what aspects does FeatureMAP significantly differ from the latest dimensionality reduction methods such as PHATE and SpaceMAP?

---

> ### Author Response · Authors · 2024-11-21
> **Thank you for your valuable suggestion. We have addressed your concerns and highlighted the revisions in pink in the manuscript.**
>
> >Weaknesses:
> 1.The paper mentions the selection of parameter λ but does not discuss the impact of other parameters on the results. Guidance on parameter selection and sensitivity analysis is crucial for practical applications.
>
> Thank you for raising this important point. In response, we have expanded in section A.1.5 (APPENDIX) of the manuscript to include the impact of key parameters $n\_neighbors$, $min\_dist$ and $\lambda$ on FeatureMAP's results. We provide guidance on parameter selection, outlining recommended default values and scenarios where adjustments may be necessary.
>
> >2.The paper should discuss the computational complexity of FeatureMAP, especially its performance and scalability when dealing with large datasets. This is a key factor for the feasibility of the method in practical applications.
>
> Thank you for your valuable suggestion. We have added a discussion on the computational complexity of FeatureMAP to address its performance and scalability with large datasets in section A.2.4 of APPENDIX.
>
> FeatureMAP’s complexity is primarily influenced by the construction of the kNN graph and the embedding process. For a dataset with $n$ samples and  $d$-dimensional input, the kNN graph construction typically has a complexity of $𝑂(n\log n)$ for approximate methods. The embedding process involves additional computations, such as embedding tangent spaces, which adds $O(nk)$, where $k$ is the size of the local neighborhood.
>
> While FeatureMAP requires more computation time than simpler methods like UMAP (e.g., 150 seconds for FeatureMAP versus 30 seconds for UMAP on 50,000 samples), its runtime is comparable to other advanced methods such as densMAP and TriMAP. These methods also involve additional steps beyond kNN graph embedding, such as density computation or triplet-based optimization.
>
> In terms of scalability, FeatureMAP is designed to handle large datasets effectively, leveraging approximate nearest neighbor search techniques to minimize computational overhead. However, as dataset sizes increase significantly (e.g., over 100,000 samples), the runtime grows due to the additional complexity of embedding tangent spaces. We have included a more detailed analysis in the manuscript to highlight these aspects and discuss practical considerations for applying FeatureMAP to large datasets.
>
> We hope this addition clarifies FeatureMAP's computational feasibility in practical applications.
>
> >Questions:
> 1.What is the computational resource consumption of FeatureMAP when dealing with large datasets? Are there optimization strategies to improve efficiency?
>
> 1.	Computational Resource Consumption of FeatureMAP for Large Datasets: FeatureMAP primarily consumes computational resources during two phases: kNN graph construction and the embedding process. For a dataset with $n$ samples and $d$-dimensional input, the resource consumption is as follows:
> -	kNN Graph Construction: Typically $O(n\log n)$ when using approximate nearest neighbor methods, which are efficient even for large datasets.
> -	Embedding Process: The embedding of tangent spaces and optimization of the manifold structure adds complexity, approximately $O(nk)$, where $k$ is the size of the neighborhood considered.
> For large datasets (e.g., over 100,000 samples), memory consumption and runtime increase due to the higher number of pairwise distances and tangent space computations. However, FeatureMAP remains comparable in computational demands to advanced methods like densMAP and TriMAP.
> 2.	We discuss the possible Optimization Strategies to Improve Efficiency:
> -	Approximate Nearest Neighbor Search: Utilize efficient libraries like Annoy or FAISS to accelerate kNN graph construction without significant accuracy loss.
> -	Dimensionality Reduction Before Embedding: Apply initial dimensionality reduction (e.g., PCA) , to lower the computational cost for tangent space embedding.
> -	Parallel Computing: Implement parallelization for distance calculations and graph-based operations to leverage multi-core CPUs or GPUs.
> -	Batch Processing: Divide the dataset into batches to reduce memory usage during embedding
> -	Adaptive Neighborhood Size: Dynamically adjust the $n\_neighbors$ parameter based on the dataset size and structure to balance runtime and embedding quality.
>
> >2 In what aspects does FeatureMAP significantly differ from the latest dimensionality reduction methods such as PHATE and SpaceMAP?
>
> Thank you for raising this question. We have included a comparative summary of FeatureMAP's advantages over other methods in Section A.1.7.
>
> Briefly:
> - PHATE: Captures temporal structures, ideal for dynamic data, but has limited feature preservation and scalability. FeatureMAP preserves features and scales efficiently.
> - SpaceMAP: Resolves crowding and preserves global geometry but lacks feature emphasis. FeatureMAP adds feature visualization and balances local-global structures.

---

> > ### Comment · Reviewer_pkDU · 2024-11-25
> >
> > Dear Authors,
> > Thank you for the effort you have invested in revising your manuscript. I have carefully reviewed the additional explanations and supplementary materials you have provided in response to the initial round of reviews. The additional experiments and data analyses strengthen the empirical foundation of your paper and provide a more comprehensive view of your method's performance and robustness.

---

> > > ### Author Response · Authors · 2024-12-01
> > >
> > > Dear Reviewer,
> > >
> > > Thank you for reviewing our revised manuscript and for your thoughtful and constructive feedback. Your comments have been instrumental in improving the quality of our work.
> > >
> > > We kindly ask if you would consider revising the score based on the changes we have made. We greatly appreciate your guidance and support throughout the review process.
> > >
> > > If you have any additional questions or suggestions, we would be delighted to address them.

---

### Official Review · Reviewer_WrKM · 2024-11-04

**Soundness:** 4
**Presentation:** 3
**Contribution:** 3
**Rating:** 6
**Confidence:** 4

**Summary:**

This paper advances manifold learning by preserving both the topological structure and source features within the tangent space, facilitating interpretability through local feature gradient analysis in the embedded tangent space. Extensive experiments in digit classification, object detection, and adversarial example analysis demonstrate the enhanced interpretability of FeatureMAP.

**Strengths:**

This paper introduces a newly designed method that provides interpretability for nonlinear dimensionality reduction. The theoretical proofs are thorough, the experiments are comprehensive, and the writing is clear.

**Weaknesses:**

Missing or incorrect punctuation following equations (e.g., Equation 11). The APPENDIX layout is somewhat unstructured.

**Questions:**

1.	Explicitly outline what current methods (e.g., UMAP) lack regarding feature preservation in the introduction, and provide a brief comparative summary to showcase why FeatureMAP fills a distinct gap.
2.	In the adversarial attack interpretation experiment (the right of Fig. 6), it is difficult to observe additional important features beyond the typical edges of digit 1. In other words, the saliency map shown is insufficient to explain the reasons for misclassification.
3.	Figure 13 shows that the computation time of FeatureMAP exceeds that of other methods. Is this time acceptable?

---

> ### Author Response · Authors · 2024-11-20
> **Thank you very much for your kind and constructive comments. We have carefully revised our paper in accordance with your valuable suggestions and have addressed all the questions you raised.**
>
> >Strengths:
> This paper introduces a newly designed method that provides interpretability for nonlinear dimensionality reduction. The theoretical proofs are thorough, the experiments are comprehensive, and the writing is clear.
>
> We thank the Reviewer for the clear summary and positive evaluation of our work. We have addressed the concerns as outlined below. The revised sections of the manuscript are highlighted in orange.
>
> >Weaknesses:
> Missing or incorrect punctuation following equations (e.g., Equation 11). The APPENDIX layout is somewhat unstructured.
>
> Thank you for pointing out these weaknesses. We have carefully reviewed and corrected the punctuation following all equations, including Equation 3, 11, 13, 14 to ensure accuracy and consistency. Additionally, we have reorganized the APPENDIX to enhance its structure and readability. The APPENDIX is now divided into two major sections: details of the algorithms and additional experimental results. Subsection titles have been added to improve clarity, and more detailed descriptions have been provided for each figure to ensure better understanding.
>
> >Questions:Explicitly outline what current methods (e.g., UMAP) lack regarding feature preservation in the introduction, and provide a brief comparative summary to showcase why FeatureMAP fills a distinct gap.
>
> Thank you for your valuable suggestion. In response, we have included a discussion in the introduction highlighting the limitations of current methods (e.g., UMAP) in feature preservation, along with a brief comparative summary. Additionally, we have provided a detailed comparative summary of major methods in a table, which is now included in the section A.1.7 of APPENDIX.
>
>
> >In the adversarial attack interpretation experiment (the right of Fig. 6), it is difficult to observe additional important features beyond the typical edges of digit 1. In other words, the saliency map shown is insufficient to explain the reasons for misclassification.
>
> Thank you for raising this question. The right panel of Fig. 6 displays five images randomly selected from the fake image dataset. In comparison with the original images and their corresponding saliency maps (Fig. 4 and the newly added Fig. 11), which highlight feature importance primarily along the edges of the digits, the saliency maps in Fig. 6 reveal additional important features beyond the digits' edges. These extra features contribute to the misclassification.
>
> As the Fast Gradient Sign Attack (FGSM) introduces small perturbations to the pixels to generate fake images, it is challenging to directly discern the misclassification from individual images. To address this, we averaged the feature importance across original and fake images to compare the saliency maps of original and adversarial labels (bottom of Fig. 6). The results indicate that the average feature importance of fake images is more similar to that of digit 8 than digit 1. This observation explains why the classifier misclassifies the fake images as digit 8.
>
> >Figure 13 shows that the computation time of FeatureMAP exceeds that of other methods. Is this time acceptable?
>
> Thank you for your question. We have expanded the description of the efficiency evaluation to provide a clearer comparison in section A.2.4 of APPENDIX.
>
> FeatureMAP does require more computation time than UMAP (e.g., 30 seconds for UMAP versus 150 seconds for FeatureMAP on 50,000 samples). However, this additional time is justified by its extended functionality. Unlike UMAP, which focuses solely on kNN graph embedding, FeatureMAP embeds tangent spaces and enables feature visualization, providing a richer representation of the data.
>
> Moreover, the runtime of FeatureMAP is comparable to densMAP and TriMAP, as all three methods involve additional computations beyond kNN graph embedding. Specifically, densMAP incorporates local density calculations, and TriMAP computes distances within triplets. In contrast, PHATE exhibits significantly higher computational overhead when the data size exceeds 50,000, due to its reliance on Multi-dimensional Scaling (MDS), which requires calculating all pairwise distances.
>
> These comparisons demonstrate that FeatureMAP's computational cost is reasonable given its enhanced capabilities and performance.

---

> ### Author Response · Authors · 2024-12-03
>
> Dear Reviewer,
>
> I hope this message finds you well. I am writing to kindly inquire about the status of the feedback for the revision based on your suggestion. I greatly value your time and expertise in reviewing our work.
>
> If there are any clarifications or additional information required from my side, please do not hesitate to let me know. Thank you very much for your efforts in providing constructive feedback.

---

### Meta-Review · Area_Chair_5v31 · 2024-12-19

**Metareview:**

While nonlinear dimensionality reduction often lacks interpretability, this paper proposes an interpretable method FeatureMAP to preserve source features by tangent space embedding. Experiments on several datasets validate the effectiveness. After the rebuttal, it receives two borderline rejects and three borderline accept. It is obviously a borderline paper. The advantages, including the thorough theoretical analysis and good presentation, are recognized by the reviewers. However, they are also concerned about the unconvincing support for better important feature preservation, unclear practical application, insufficient convergence analysis, etc. I agree with Reviewer 57HL that the proposed method lacks clear criteria for comparison and judgment. The results presented in Table 1 do not show the superiority, where they are very close. Moreover, the datasets are relatively small from my point of view. The large-scale datasets application is not validated. I think the current manuscript does not meet the requirements of this top conference. I suggest the authors carefully revise the paper and submit it to another relevant venue.

**Additional Comments On Reviewer Discussion:**

The response well addresses some concerns. Reviewer 57HL raises the score but still does not suggest acceptance in the discussion. This is a borderline paper. There is still much room for improvement from my point of view, such as the experimental results comparison, larger dataset, etc.

---

### Decision · Program_Chairs · 2025-01-22

Reject